# Active Constituent of HQS in T2DM Intervention: Efficacy and Mechanistic Insights

**DOI:** 10.3390/ijms26104578

**Published:** 2025-05-10

**Authors:** Yaping Chen, Qiuqi Wen, Bing Yang, Liang Feng, Xiaobin Jia

**Affiliations:** School of Traditional Chinese Pharmacy, Innovation Center for Industry-Education Integration of Pediatrics and Traditional Chinese Medicine, State Key Laboratory of Natural Medicines, China Pharmaceutical University, Nanjing 211198, China; 13382059050@163.com (Y.C.);

**Keywords:** active constituent, Huang-Qi San, type 2 diabetes, potential pharmacodynamic ingredients

## Abstract

Traditional Chinese Medicine (TCM) is recognized for its complex composition and multiple therapeutic targets. Current pharmacological research often concentrates on extracts or individual components. The former approach faces numerous challenges, whereas the latter oversimplifies and disregards the synergistic effects of TCM components. This study aimed to address this limitation by evaluating the therapeutic efficacy and mechanisms of Huang-Qi San (HQS) active constituent (AC) against type 2 diabetes (T2DM). Active components of HQS were identified using network pharmacology and spectrum–effect correlation analysis. The reconstituted AC group was assessed both in vitro (for glucose consumption and glycogen synthesis) and in vivo (in T2DM mice), with metabolomics and molecular docking techniques used to elucidate the underlying mechanisms. Eight components exhibiting a correlation degree greater than 0.85 were identified as the representative components of HQS intervention for T2DM. These eight components were then mixed in equal proportions to produce AC. The AC group demonstrated increased glucose uptake and glycogen synthesis in vitro, surpassing both the HQS extract and individual components. In diabetic mice, AC significantly increased the insulin sensitivity, outperforming the HQS extract and matching the efficacy of metformin. Metabolomics analysis identified pentose and glucuronic acid interconversion as a critical metabolic pathway, with strong binding affinity (less than −15 kJ/mol) between AC and key enzymes. This research further substantiates the scientific validity and feasibility of emphasizing active constituents in the evaluation of TCM efficacy. Additionally, it provides a scientific foundation for the clinical application of HQS. Most importantly, this study serves as a demonstration of the development of new TCM drugs characterized by clear ingredients, safety, and effectiveness.

## 1. Introduction

Diabetes mellitus (DM) is a complicated and serious metabolic disorder characterized by abnormally high blood glucose levels that are due to impaired insulin secretion or action [1]. As a typical metabolic disease, type 2 diabetes mellitus (T2DM) often involves complex metabolic pathways, which pose challenges in investigating associated pathophysiological mechanisms and therapeutic interventions [2,3]. Most chemical drugs can entail significant adverse effects while exhibiting potent anti-T2DM activity [4].

Numerous studies have demonstrated that Traditional Chinese Medicine exhibits fewer adverse effects while ameliorating T2DM. Huang-Qi San (HQS), originating from ‘Sheng Ji Zong Lu’ during the Song Dynasty for the treatment of diabetes mellitus, consists of three components (*Astragalus membranaceus*, *Pueraria lobata*, *Morus alba* L.) in a proportion of 1:2:1. Studies have revealed that Astragali Radix, Puerariae Radix, and Cortex Mori Radicis exhibit beneficial influences on diabetes and its complications, both individually and synergistically [5]. Additionally, HQS may have certain effects in improving blood circulation, lowering blood pressure, and reducing swelling, with no reported side effects. However, only a limited number of studies have examined the effects of HQS on T2DM [6]. The active compounds and mechanisms underlying the use of HQS as an intervention for T2DM remain elusive. Consequently, it is imperative to undertake an extensive and systematic exploration of metabolic regulation in T2DM following intervention with HQS to furnish a scientific foundation for its clinical utilization.

Recent studies suggest that the effectiveness of Traditional Chinese Medicine (TCM) largely arises from the collective synergistic impacts of its various components and their systematic modulation of numerous targets. Nevertheless, current research into the pharmacological agents of TCM mainly focuses on isolated individual compounds or extracts [7,8]. Examining extracts poses significant difficulties because of the intricate nature of their individual components. In contrast, studies focused exclusively on one particular element often simplify the matter excessively, disregarding the comprehensive and synergistic interactions among various components in TCM, which restricts the potential therapeutic efficacy of the drug. Therefore, we proposed that the active constituent (AC) resulting from the combination of several active compounds could be central to efficacy investigations in TCM. This strategy not only alleviates the hurdles related to examining the effective substances in TCM but also considers the synergistic influences inherent in its multiple components. Our research team has carried out pertinent studies demonstrating that prioritizing AC as the focal point in Traditional Chinese Medicine (TCM) is both rational and practical [9]. Hence, this research intends to adopt this framework to assess and validate the effects of AC of HQS in relation to T2DM. We hope to develop a new TCM drug with more defined components compared to the HQS extract, which is also safer and possibly more effective than metformin.

In this research, the disease model utilized was T2DM to explore the active constituent of HQS in relation to T2DM intervention. First, the potential active ingredients of the HQS extract for treating T2DM were identified by integrating network pharmacology with spectrum–effect correlation analysis. Subsequently, various identified potential active components were administered either alone or in combination to evaluate their therapeutic impacts on T2DM. Furthermore, we will investigate the possible mechanisms of action using non-targeted metabolomics technology and molecular docking, thereby providing a scientific basis for the clinical application of HQS.

## 2. Results

### 2.1. Network Pharmacology

#### 2.1.1. Gene Collection

A detailed compilation of T2DM targets resulted in the identification of 16,984 targets retrieved from various databases. The chemical properties of 39 active compounds were confirmed via the PubChem database. Next, the duplicates were removed and we then found 219 disease targets affected by HQS. By intersecting this set of 219 predicted target proteins with the 16,984 disease-related genes, we successfully identified the potential targets of HQS relevant to T2DM management. This target information was further verified through the UniProt database. In conclusion, we recognized 155 targets for HQS linked to the treatment of T2DM, as shown in Figure 1A. A direct interaction network was developed using Cytoscape to clarify the relationships between HQS-active ingredients and target proteins, as illustrated in Figure 1B.

#### 2.1.2. PPI Network Visualization and Key Target Discovery

This research involved an extensive examination of 155 genes linked to the treatment of T2DM through the herbal formulation HQS. These genes were represented via a PPI network, which was then visualized using the Cytoscape. The resulting network representation is illustrated in Figure 1C. To analyze the network’s characteristics, the CytoNCA algorithm was utilized, focusing on key metrics such as degree, betweenness, and closeness. This analytical approach provided insights into the network’s topology, which was evaluated using Cytoscape version 3.9.1. As detailed in Table 1, out of the numerous components analyzed, 28 were identified that exhibited a degree greater than zero. This finding suggests that these 28 compounds could potentially serve as the active ingredients responsible for the beneficial effects of HQS in the management of T2DM. In light of this discovery, a spectral-effect correlation analysis was subsequently conducted on these 28 components. This step aimed to further refine the selection of representative ingredients with significant potential for intervention in the context of type 2 diabetes.

### 2.2. Spectrum–Effect Correlation Analysis

The analysis and identification of the various compounds that were absorbed into the bloodstream of rats in the HQS group were carried out on UPLC-Q-TOF-MS. The detailed information regarding these identified components has been systematically documented and is presented in Table 2. To further explore the relationship between these components and different pharmacological indicators, gray correlation analysis was undertaken. The results, illustrated in Figure 2, reveal that eight components exhibited a correlation degree exceeding 0.85. This significant correlation indicates that these eight components are likely to be potential active compounds of HQS in the intervention of T2DM.

### 2.3. Quantitative Analysis of 8 Active Components of HQS

The validated UPLC method was employed for the quantification of eight representative components in the HQS-effective extract samples. Each analyte was quantified using its respective calibration curve, and results from three independent determinations demonstrated that all eight components were detected across all of the HQS-effective extract samples. The results of the content measurements are presented in Table 3 and Figure 3.

### 2.4. In Vitro Pharmacological Study of HQS in Alleviating T2DM

The findings from the network pharmacology and spectrum–effect correlation analysis indicated that eight components, namely Puerarin, Formononetin, Astragaloside IV, Astragaloside III, Kaempferol, Tiliroside, Scopoletin, and Moracin A, were examined further as possible active ingredients for the treatment of T2DM using HQS.

#### 2.4.1. Cytotoxicity Study

The effects of different drugs on the growth and proliferation of HepG2 cells were observed by CCK8. This assay is a reliable method for measuring cell viability and proliferation in response to the treatment. Additionally, the concentration ranges that were deemed safe for each of the drug components tested have been clearly illustrated in Figure 4.

#### 2.4.2. The Hypoglycemic Effect on IR-HepG2 Cells

To induce insulin resistance (IR) in HepG2 cells, a solution containing insulin (INS, 0.06 IU/mL) was administered for a duration of 24 h. The model is considered successful if the glucose consumption of HepG2 cells subjected to INS intervention is significantly lower than that of cells without insulin intervention [10,11]. As illustrated in Figure 5, insulin stimulation led to a notable reduction in glucose consumption by HepG2 cells, thereby confirming the successful establishment of the IR-HepG2 cell model. The next step involved administering the culture medium supplemented with various concentrations of HQS extract, eight components, active constituent, and metformin (20 μM). After another 24 h of incubation, the glucose content was measured using an automatic biochemical analyzer. The specific outcomes of this experiment are illustrated in Figure 5A–J. According to this result, the effective concentrations of each drug was confirmed as follows: Astragaloside IV (160 μM), Astragaloside III (90 μM), Kaempferol (5 μM), Tiliroside (5 μM), Formononetin (20 μM), Scopoletin (25 μM), Moracin A (70 μM), Puerarin (20 μM), HQS extract (25 μg/mL), and AC (20 μg/mL). This established a framework for subsequent assessments of glycogen level and glucose consumption in the IR-HepG2 cells, which were evaluated at these specified effective concentrations for HQS, as well as for the eight active components and AC. The findings, displayed in Figure 5K,L, demonstrate that HQS significantly enhances glucose utilization and promotes glycogen production in IR-HepG2 cells. Results demonstrated that all the dosing groups had significant effects, and the AC exhibits an effect akin to that of the Met.

### 2.5. In Vivo Pharmacological Study of HQS in Alleviating T2DM

#### 2.5.1. Hypoglycemic Effect of HQS on T2DM Mice

The administration of streptozotocin (STZ) at a dosage of 40 mg/kg, in conjunction with a diet rich in fats and sugars, may result in elevated levels of fasting blood glucose (FBG). After a treatment period of eight weeks, the administration of HQS was found to significantly reduce both the FBG levels and the food intake in mice, as demonstrated in Figure 6A–C. After 8 weeks of administration, the FBG levels in each treatment group decreased significantly by approximately 50% to 58% compared to the model group. Importantly, the active constituent displayed analogous effects that were dependent on the dosage, indicating a clear relationship between the amount administered and the resultant physiological changes.

The findings from the Oral Glucose Tolerance Test (OGTT) reveal that throughout the 180 min observation period, every group peaked in blood glucose levels at the 30 min point (Figure 6D). This observation illustrates the common pattern of blood glucose levels, which first increase and then decrease. Importantly, the blood glucose levels were significantly higher in the Model, indicating a substantial impairment in glucose tolerance in the diabetes-affected mice. In contrast, the Control showed a full restoration of blood glucose levels to baseline after 180 min post-oral glucose administration. Furthermore, an assessment of the treatment groups, including the Met, HQS, and AC groups, revealed that their blood glucose levels were considerably lower than those in the Model. After the 180 min period, these groups also normalized their blood glucose levels back to initial values. This finding is consistent with the area under the curve (AUC) calculations depicted in Figure 6E. The Model exhibited an approximate increase of 2.8 times compared to the Control, while Met and AC_H represented approximately 79% and 72% of the Model. Among the treatment options, the AC group demonstrated particularly pronounced effects on the OGTT results in T2DM mice, underscoring their potential as therapeutic strategies for addressing glucose intolerance.

#### 2.5.2. Improvement of Insulin Sensitivity in T2DM Mice

The Model displayed significantly elevated levels of INS when contrasted with the Control (Figure 7A). After the administration of metformin and HQS, INS levels in the mice that underwent these treatments—specifically in the Met and AC_H group—exhibited a significant reduction (*p* < 0.01). In these groups, both HQS and AC treatments demonstrated significant therapeutic effects, indicating their efficacy in regulating INS levels. Furthermore, assessments of HOMA-IR and HOMA-β were conducted (Figure 7B,C). The findings indicated a significant increase in HOMA-IR, approximately 4 times that of the Control, alongside a decrease in HOMA-β, which was about 31% of the Control, for the Model. However, following treatment with HQS and Met, a reversal of these parameters was observed in diabetes, particularly in the AC_H group, where HOMA-IR was reduced to half that of the Model, and HOMA-β was approximately twice that of the Model. This suggests that HQS can effectively counteract the damage inflicted on pancreatic islet β cells, thereby restoring their functionality. It is worth mentioning that the AC group produced notably better results in this respect.

The results of insulin tolerance test (ITT) were as illustrated in Figure 7D,E, showing that mice with diabetes that received treatments involving HQS and metformin showed markedly greater decreases in blood glucose levels compared to their untreated diabetic counterparts. This result highlights the possible therapeutic advantages of both HQS and metformin for managing glucose metabolism. Furthermore, a quantitative evaluation for the ITT provided additional validation for these findings. The analysis demonstrated that the AUC_ITT_ for the Model was notably higher compared to the Control (*p* < 0.01, approximately 3.4 times), indicating a diminished insulin response in T2DM. In contrast, the AUC_ITT_ values for the Met, HQS, and AC groups were significantly lower than those recorded in the Model (*p* < 0.01, approximately 63%~68% of the Model), with AC groups exhibiting effects comparable to Met. This implies that diabetic mice given HQS and the active constituent showed improved insulin sensitivity. Overall, these results suggest that both HQS and the AC could enhance insulin sensitivity in mice with diabetes and facilitate the regeneration of impaired islet β cells.

#### 2.5.3. Effects of HQS on Serum Biochemical Indices of T2DM

To examine how HQS affects the serum indices associated with T2DM, we performed a thorough analysis of the critical parameters. In this study, we conducted a detailed analysis of various lipid and metabolic parameters by measuring the concentrations of high-density lipoprotein cholesterol (HDL-C), low-density lipoprotein cholesterol (LDL-C), total cholesterol (T-CHO), triglycerides (TGs), glycated hemoglobin (GHb), and ketone bodies (KB). The results demonstrate a statistically significant rise (*p* < 0.01) in the levels of GHb, LDL-C, T-CHO, and TG within the model group (Figure 8). Conversely, the levels of KB and HDL-C were notably reduced (*p* < 0.01) in the model group. Following the administration of HQS and AC, we observed that the previously altered biochemical indicators returned to normal levels. These outcomes suggest that both HQS and AC possess the potential to ameliorate the glycolipid metabolic disturbances commonly associated with T2DM in mice, akin to the effect of metformin, a well-known anti-diabetic medication. Notably, the active component displayed superior efficacy than HQS in regulating glycolipid metabolic disturbances.

#### 2.5.4. Pathological Changes in Liver and Pancreas Tissue in T2DM Mice

The histological examination results revealed that both HQS and its active component exhibited a significant improvement in the morphological changes in the liver and pancreas (Figure 9). Notably, we observed pronounced pathological alterations, including the presence of inflammatory cell infiltration, fat vacuoles, severe damage to liver morphology, and hepatocyte necrosis. Additionally, the study observed a significant decrease in both the quantity and size of pancreatic islets in the pancreas of T2DM mice. Accompanying these changes, there were notable pathological features such as turbidity, swelling, and vacuolar degeneration of the pancreatic islet cells. Furthermore, an infiltration of inflammatory cells was also present within the pancreatic tissue, indicating an adverse response linked to the disease. However, the administration of HQS and AC demonstrated a significant efficacy in mitigating these morphological changes. The treatment appeared to restore the structure and function of the pancreatic islets, suggesting a potential therapeutic strategy for alleviating some of the detrimental effects of T2DM. Furthermore, PAS staining was performed to assess glycogen distribution within the liver. Under conditions of insulin resistance, glycogen synthesis is hindered, resulting in decreased glycogen levels and elevated blood glucose. The intensity of the purple coloration observed after PAS staining correlates with glycogen content. In Figure 9B, it is evident that the mice belonging to the Model displayed a significant reduction in liver glycogen levels. This marked depletion suggests that there may be underlying metabolic alterations in the Model that affect glycogen storage and utilization, highlighting the potential physiological implications of the conditions being studied. Yet, after an eight-week treatment period, the accumulation of liver glycogen in the HQS and AC groups significantly increased in a concentration-dependent manner, indicating that HQS and AC effectively mitigated the liver glycogen depletion resulting from type 2 diabetes.

### 2.6. Metabolomics Analysis

#### 2.6.1. Multivariate Statistical Analysis

Multivariate statistical analysis was conducted on the UPLC-Q-TOF/MS data to further elucidate the metabolic differences among the groups. Firstly, samples from each group underwent an unsupervised PCA, with the results depicted in Figure 10A. The validity of this approach was evidenced by the high aggregation of QC samples in the negative ion mode. The variables identified in the samples were verified to hold physiological significance within a 95% confidence interval. Additionally, it was observed that the samples from the same group tended to cluster closely together, indicating a strong correlation among them. To further evaluate these relationships, PLS-DA was applied to the rat plasma samples. This analysis showed that there was a notable clustering effect among the four distinct groups, reinforcing the validity of the observed relationships and suggesting clear differentiation among the sample groups based on the variables assessed. This suggests that HQS exerts a notable effect on the metabolic abnormalities in rats caused by T2DM, as illustrated in Figure 10B.

To delve deeper into the therapeutic mechanism underlying HQS, we conducted an OPLS-DA analysis aimed at identifying ions that exhibited significant variations across the Control, Model, and HQS groups. The results, displayed in Figure 10C,D, clearly illustrate that the Control, the Model, and the HQS were distinctly separated from one another when analyzed in negative ion mode. This distinction reveals significant variations in the ionic characteristics of the groups, potentially offering insights into the therapeutic impacts of HQS. In the case of the Control and Model groups, R^2^Y = 0.998 and Q^2^ = 0.927 were observed in negative ion mode. R^2^Y = 0.996 and Q^2^ = 0.963 were observed in negative ion mode for the Model and HQS groups. The findings indicated that the Model showed robust fitting and predictive abilities, making it appropriate for identifying differential metabolites.

Volcano plots illustrate trends in differential metabolite expressions between the Control and Model groups (Figure 10E), as well as between the Model and HQS groups (Figure 10F). Every point displayed on the graph represents a particular metabolite. The area to the left of the origin indicates metabolites that are down-regulated, while the area to the right signifies those that are up-regulated.

#### 2.6.2. Identification of Potential Biomarkers

In the analysis conducted using OPLS-DA models, significant results were obtained with a VIP ≥ 1 and a *p*-value ≤ 0.05. These thresholds allowed for the identification of a total of 28 differential metabolites when comparing the model and control groups. Furthermore, when examining the model and HQS groups, 44 differential metabolites were identified. In addition, a Fold Change greater than 2 and an absolute value of |*p*(corr)| greater than 0.6 were also applied as criteria for metabolite differentiation, reinforcing the robustness of the findings. Venny analysis was employed to screen for differential metabolites among the three groups, and 18 metabolites were selected (Figure 11A and Table 4). Among these, two types of differential metabolites showed a significant increase in the Model group but decreased following treatment. Conversely, the remaining 16 differential metabolites exhibited the opposite trend. The databases HMDB, METLIN, KEGG, and others were employed to identify, annotate, and determine differential metabolites exhibiting a high accuracy of ligand ion binding within a margin of ±10 ppm.

Heat map analysis was conducted on the differential metabolites in the serum. The results indicated significant differences in metabolite levels between the normal and T2DM rats, with the HQS group showing a tendency to restore metabolite levels to those observed in the Control. This suggests that HQS may have the potential to correct the abnormal serum metabolite levels in T2DM rats (Figure 11B).

#### 2.6.3. Metabolic Pathway Analysis

The MetaboAnalyst 5.0 database was employed to analyze the metabolic pathways associated with the common differential metabolites across the three groups. The resulting metabolic pathways are illustrated in Figure 12A–C. The figure presents a clear representation of various metabolic pathways, with each circle serving as a graphical representation of a specific pathway. The coloration of these circles reflects the −log (*p*-value), which is detailed on the vertical axis of the figure. Meanwhile, the horizontal axis illustrates the enrichment factor associated with each pathway. A greater enrichment factor indicates a more substantial presence of differential metabolites within that particular pathway. Notably, the pathway located in the upper right corner of the figure is highlighted as the one most significantly affected, suggesting it experiences the highest level of influence in relation to the differential metabolites analyzed.

A metabolic pathway analysis conducted using MetaboAnalyst revealed that potential biomarkers are involved in three main pathological processes disrupted in T2DM: pentose and glucuronate interconversion pathway, glycerophospholipid metabolism, and fatty acid biosynthesis. These findings (Table 5, Figure 12D) suggest that the anti-diabetic effect of HQS is primarily due to its modulation of glycometabolism and lipometabolism. Notably, the pentose and glucuronate interconversions pathway was significantly disrupted (*p*-value < 0.05, impact = 0.14062), prompting plans for subsequent experiments to validate this pathway.

### 2.7. Molecular Docking

To elucidate the relationship between HQS and the interconversion pathway of pentose and glucuronic acid, molecular docking studies were conducted involving eight representative components of HQS and four crucial enzymes of the pathway (UGT1A1, UGT1A9, UGT2B10, and UGT2B15). The binding energies were assessed to evaluate the docking results, with values below zero indicating spontaneous binding of the ligand to the receptor. A binding energy < −5 kJ/mol is considered indicative of a significant binding affinity of a compound for the target protein [12]. As shown in Table 6, all binding energies were below −10 kJ/mol, suggesting that all eight representative components form stable complexes with the four enzymes. Notably, the binding energies for the interactions between the compounds and UGT2B15 and UGT2B10 were recorded as less than −19 kJ/mol, indicating a stronger affinity for these enzymes. To visualize the interactions of the compounds with their respective targets, the docking results are presented in Figure 13. Hydrogen bonds represent the primary intermolecular forces facilitating the binding between the compounds and the targets.

## 3. Discussion

Diabetes is a chronic condition characterized by significant metabolic disturbances in carbohydrate and lipid metabolism, arising from insulin deficiency or dysfunction [13,14]. As time progresses, these metabolic irregularities may result in multiple complications, positioning T2DM as a significant public health issue that necessitates diligent management and intervention [15]. Therefore, it is effectively crucial for the treatment of T2DM to manage hyperglycemia and hyperlipidemia. As a traditional prescription known for its hypoglycemic effects, HQS is widely utilized in China. This study screened the active constituent of HQS that alleviates T2DM and verified their efficacy both in vivo and in vitro, providing novel insights and a new research paradigm for exploring the active ingredients in TCM.

In the realm of network pharmacology, a detailed examination identified 219 direct target genes linked to 39 active compounds present in HQS. This discovery emphasizes the considerable pharmacological capabilities of HQS in addressing T2DM. The network pharmacology findings revealed that the degree values of 28 components are all above zero, indicating a stronger association with the primary target. Consequently, these 28 components can be regarded as the active elements of HQS in relieving T2DM. Subsequently, a spectrum–effect correlation analysis was conducted based on these 28 components to further elucidate the representative components of HQS intervention in T2DM.

The correlation between spectrum and effect involves performing a statistical analysis to evaluate the relationship between the chemical constituents of TCM and their corresponding therapeutic outcomes [16]. This research utilized gray relational analysis to examine how the blood-entering components of HQS extract relate to its effectiveness in enhancing T2DM indicators in rats. Using a correlation degree threshold of 0.85 as the evaluation standard [17], eight active components were identified: Puerarin, Formononetin, Astragaloside IV, Astragaloside III, Kaempferol, Tiliroside, Scopoletin, and Moracin A. These components are closely associated with T2DM. In the subsequent content determination experiment, the presence of these eight active ingredients in the HQS extract was also confirmed. To investigate the effects of these active ingredients, the eight components were combined in equal proportions to create a mixture representing the active constituent of HQS, followed by pharmacodynamic studies conducted both in vivo and in vitro.

Insulin resistance (IR) plays a pivotal role in the onset of T2DM, along with metabolic abnormalities, such as high FBG, dyslipidemia, and heightened release of inflammatory substances [18]. In the in vitro pharmacodynamic study, a model using IR-HepG2 cells was created based on glucose consumption to evaluate the effective concentration of the extract from HQS, alongside eight active components and the AC. After drug administration, there was a notable increase in glucose uptake and utilization. This observation implies that HQS extract, alongside the eight active compounds and the AC, significantly improve glucose utilization in IR-HepG2 cells. Previous studies have indicated that insulin resistance chiefly results in disrupted glycogen synthesis and impaired regulation of glucose production in hepatocytes [19]. The findings indicated that IR-HepG2 cells exhibited notably lower amounts of intracellular glycogen. Nevertheless, the administration of HQS extract, along with its eight active ingredients and the AC, resulted in an enhancement of glycogen levels within these cells. This finding suggests that HQS can enhance glycogen synthesis. Notably, the AC showed greater efficacy compared to the HQS extract and the eight active ingredients when evaluated separately, demonstrating effects similar to those of the Met.

Pharmacodynamic studies in vivo were performed utilizing a mouse model of T2DM induced by STZ. Over the course of eight weeks, we noted changes in the dietary intake and FBG of the mice [20,21]. After administering the HQS extract along with its active component, treated mice exhibited a decrease in both food and water consumption when contrasted with the diabetic control group, suggesting an improvement in these symptoms. The findings showed that the active component of HQS significantly lowered levels of the FBG in a dose-dependent fashion.

To investigate the effect of HQS on insulin sensitivity in mice with T2DM, insulin resistance in each group of mice was assessed through HOMA-IR and HOMA-β [22]. Following treatment with the extract of HQS and AC, an increase in HOMA-β and a decrease in HOMA-IR were noted, suggesting an improvement in the symptoms of IR. The results of ITT revealed that both the HQS extract and AC significantly boost INS sensitivity in T2DM. The findings indicate that both the HQS extract and AC could effectively reduce blood glucose levels by influencing IR and insulin tolerance. Importantly, the administration of high doses of AC demonstrated more pronounced effects than the HQS extract alone, with the active constituent showing efficacy that is on par with metformin.

The metabolism of glycolipids in the human organism is interconnected; when there is a dysfunction in glucose metabolism, it negatively influences lipid metabolism as well [23]. Consequently, we assessed the blood lipid profile in the mice with T2DM in this study, with a particular focus on the concentrations of TG, T-CHO, HDL-C, and LDL-C. The results indicate that treatment with HQS extract and AC significantly reduced blood lipid levels in T2DM mice, while HDL-C levels increased. This finding is corroborated by supporting evidence. Although the lipid-regulating ability of the AC in T2DM mice was slightly less effective than that of Met, its effect still surpassed that of the HQS extract.

The liver is essential for glucolipid metabolism. Injury to the liver can lead to dyslipidemia, often presenting as IR and difficulties in glycogen production within the liver [24]. The analysis of pathological changes in the pancreas and liver of this model utilized H&E staining, highlighting an enhancement in these lesions after treatment. Findings from the study indicate that the extract of HQS and AC might enhance the metabolism of glucose and lipids by promoting the healing of liver cells. Moreover, the liver serves as an essential hub for the production and storage of glycogen, contributing significantly to the maintenance of overall metabolic balance. The PAS staining was employed to evaluate how HQS extract affects liver glycogen concentrations. Findings from this analysis indicate that glycogen accumulation in the liver markedly increased in the livers of T2DM mice that received HQS extract and AC. This observation emphasizes the effectiveness of AC in enhancing hepatic glycogen production, thereby contributing to the regulation of FBG. The findings emphasize the potential therapeutic benefits of HQS extract and AC in the treatment of metabolic disorders, especially in hyperglycemia controlling.

Previous pharmacodynamic studies have demonstrated that the AC derived from a balanced blend of eight active compounds in HQS can significantly alleviate T2DM. This active constituent has shown greater efficacy compared to both the HQS extract and the individual administration of the eight ingredients, and it is equally effective as metformin in lowering blood glucose levels. These findings underscore the importance of focusing on the active ingredient in TCM therapeutic research. Subsequently, metabolomics analysis was employed to identify endogenous metabolites associated with HQS treatment of T2DM. The metabolite profiles of both the Model and HQS groups were established, and correlations were analyzed. The results indicated that the metabolic disorders in T2DM rats were primarily linked to pentose–glucuronate interconversions, drug metabolism via cytochrome P450, unsaturated fatty acid biosynthesis, glycerophospholipid metabolism, as well as fatty acid elongation, degradation, and biosynthesis. Among them, pentose-glucuronate interconversions have the most significant changes (*p*-value < 0.05, impact = 0.14062). In the results of potential biomarker identification, the concentration of lithocholate 3-O-glucuronide was significantly reduced in T2DM rats but was restored to normal levels following HQS intervention. Lithocholic acid, a secondary bile acid, is intimately linked to diabetes and glycolipid metabolism [25]. Research has demonstrated that UDP-glucuronate is reversibly converted into lithocholate 3-O-glucuronide and UDP-glucose through the action of corresponding enzymes, implicating the regulation of glucuronosyltransferase activity [26]. The decrease in the concentration of lithocholate 3-O-glucuronide indicates that the activity of the pentose and glucuronate interconversions pathway in T2DM is diminished. The reduced activity of the pathway results in a decreased synthesis of UDP-glucuronic acid (UDPGA), which is a crucial molecule for liver detoxification. A deficiency in UDPGA obstructs the metabolism of exogenous drugs, leading to the accumulation of toxic metabolites, including advanced glycation end products (AGEs). This accumulation can directly damage pancreatic β cells, exacerbate insulin resistance, and finally result in diabetes. To further elucidate the mechanisms underlying HQS’s improvement of T2DM, we will investigate the impacts of HQS on the pentose and glucuronate interconversions pathway in T2DM rats through molecular docking.

The affinity between eight representative components of HQS and the key proteins UDP glycosyltransferase (UGT) in the pentose and glucuronate interconversions pathway was predicted by molecular docking. The UGT superfamily, which is characterized by similarities in amino acid sequences, is broadly categorized into two principal families: UGT1 and UGT2. The UGT1 gene is notable for containing a distinct initial exon that encodes the N-terminal domain. This exon then connects with the common exons 2 to 5, ultimately resulting in the formation of the C-terminal domain. In contrast, the isoforms of UGT2 are encoded by separate genes that include a total of six unique exons. Furthermore, the UGT2 family is further classified into two subfamilies: UGT2A and UGT2B. Molecular docking results suggested that the eight components interact strongly with both UGT1 and UGT2, with even stronger interactions observed with UGT2 (binding energy < −19 kJ/mol), indicating that these compounds may affect pentose and glucuronic acid interconversion. Thus, the pentose and glucuronic acid interconversion pathway may represent a key mechanism through which HQS interferes with T2DM.

## 4. Materials and Methods

### 4.1. Materials

*Astragalus membranaceus* (Fisch.) Bunge, *Pueraria lobata* (Willd.) Ohwi, and *Morus alba* L. were obtained from the Bozhou Baopu Pharmaceutical Co., Ltd. (Bozhou, China). All samples were authenticated by Long Wang (School of Traditional Chinese Pharmacy, China Pharmaceutical University, China). Puerarin (B20446, purity ≥ 98%) was purchased from Shanghai Yuanye Biotechnology Co., Ltd. (Shanghai, China). Formononetin (PS000674, purity ≥ 98%), Tiliroside (PS010822, purity ≥ 98%), Moracin A (PS010299, purity ≥ 98%), Scopoletin (PS010525, purity ≥ 98%), Astragaloside IV (PS012327, purity ≥ 98%), Kaempferol (PS011599, purity ≥ 98%), and Astragaloside III (PS020942, purity ≥ 98%) were purchased from Push Bio-technology Co., Ltd. (Chengdu, China). Streptozocin (STZ, C1616118, with a purity of ≥ 98%) and Metformin hydrochloride (Met, I1904088, also with a purity of ≥98%) were sourced from Shanghai Aladdin Bio-Chem Technology Co., Ltd. (Shanghai, China). Sodium citrate and citric acid were acquired from Sinopharm Chemical Reagent Co., Ltd., which is also based in Shanghai, China. ACCU-CHEK Active Blood Glucose Test Strips were procured from Roche Diagnostics Co., Ltd., (Shanghai, China). The high-fat, high-sugar diet, which contains 24.73% protein (providing 19.05% of total calories), 34.33% fat (contributing 60.39% of total calories), and 26.29% carbohydrates (accounting for 20.56% of total calories), was obtained from Jiangsu Synergy Medicine Bioengineering Co., Ltd. (Nanjing, China). Kits for TG, T-CHO, LDL-C, and HDL-C were sourced from Nanjing Jiancheng Bioengineering Institute in Nanjing, China. Additionally, ELISA kits for INS, GHb, and KB were acquired from Jiangsu Meimian Industrial Co., Ltd. in Rugao, China.

### 4.2. Network Pharmacology

#### 4.2.1. Gene Collection of T2DM

Genes that are linked to T2DM were gathered from a variety of reputable databases. The sources utilized for this purpose included the OMIM [27], GeneCards [28], PharmGKB [29], and DrugBank [30]. Through these comprehensive resources, relevant genes associated with T2DM were identified, allowing for a targeted selection of pertinent genetic information related to the condition.

#### 4.2.2. Prediction and Identification of HQS Components and Potential Targets

The information of the compounds (including chemical name, molecular formula, PubChem Compound Identifier (CID), and Canonical SMILES) was systematically gathered from the PubChem database (https://pubchem.ncbi.nlm.nih.gov/, accessed on 14 January 2022). This comprehensive collection allows for a thorough understanding of the individual compounds present in the study. Enhancing the prediction of targets related to the active components of these elements involved the use of the TCMSP. This approach successfully identified the target profiles associated with the effective components of HQS, specifically including Puerariae Radix, Astragali Radix, and Cortex Mori Radicis. A comparative analysis was conducted between the identified component target and the disease gene set. This step was crucial for pinpointing the possible target set of HQS that could be effective for T2DM treatment. The target information was further validated through the UniProt database, accessible at https://www.uniprot.org/, accessed on 16 January 2022, ensuring the accuracy and reliability of the findings. In summary, the data concerning the components and targets related to HQS for the treatment of T2DM were meticulously compiled. An active ingredient–target network was constructed utilizing Cytoscape (V 3.7.2) to illustrate the relationship between HQS for T2DM treatment.

#### 4.2.3. Protein–Protein Interaction (PPI) Network Analysis

This study performed a comprehensive assessment of key targets within the treatment network for T2DM utilizing HQS, aiming to improve the investigation and identify the main targets by analyzing the modules of the PPI network. The study employed CytoNCA tools, which are incorporated into the Cytoscape software (V 3.7.2), to analyze various metrics of network topology relevant to the identified targets. Among these metrics are degree, betweenness, and closeness, each of which plays a pivotal role in understanding both the structural characteristics and dynamic behaviors of the network. These metrics help elucidate how different targets interact within the network, providing valuable insights into their potential effectiveness in T2DM treatment.

### 4.3. Preparation of the HQS Extract

A combination of 600 g of Puerariae Radix, 300 g of Astragali Radix, and 300 g of Cortex Mori Radicis was ground and mixed with an aqueous solution of 80% ethanol in a 1:10 (g/mL) ratio. The mixture obtained was subjected to reflux extraction at a temperature of 95 ± 2 °C for a period of 2 h, after which filtration was performed to obtain the filtrate. This extraction process was repeated, and the filtrate was combined and underwent further treatment through vacuum-rotary evaporation (65 °C, −0.1 MPa) and freeze-drying. This series of techniques led to the successful extraction of 213 g of the final product obtained from HQS.

### 4.4. Spectrum–Effect Correlation Analysis

#### 4.4.1. Animal Models

A total of 24 male SD rats, each with a weight range of 180 to 220 g, were procured from the Qinglongshan Animal Breeding Farm, which operates under the license number SCXK (Zhe) 2021-0192 in China. These animals were subsequently housed in the specialized barrier facility located at the New Animal Center of China Pharmaceutical University. The environmental conditions were maintained at 23 ± 2 °C with a relative humidity of 45 ± 10%. The rats were kept on a 12 h light–dark cycle and had unrestricted access to water and food. Approval for all procedures was secured from the Experimental Animal Ethics and Management Committee at the China Pharmaceutical University, ensuring adherence to applicable international, national, and institutional regulations concerning the management and care of animals. This experiment was conducted meticulously in accordance with international animal welfare guidelines. The designated identification number for this study is 2021-12-026.

The rats were randomly assigned to 4 distinct groups (n = 6): the control group (Control), the model group (Model), the group treated with HQS extract (1.2 g/kg/day), and the positive control group receiving metformin (Met, 200 mg/kg/day), with the solvent being water. The control group was provided with standard feed, while the other groups were subjected to a high-fat and high-sugar diet to accelerate the development of T2DM. The rats were fed for four weeks and subsequently underwent a 12 h period of fasting. Both the model group and the HQS group received STZ (40 mg/kg, i.p.) in an ice bath within a three-day period. The STZ was dissolved in a sodium citrate buffer solution (0.1 mol/L, pH 4.4). In contrast, the control group was injected with an equivalent amount of sodium citrate buffer. The diagnosis of diabetes in the rats was based on the presence of hyperglycemia, characterized by FBG ≥ 11.1 mmol/L. Following this, the rats administered water in the control and model groups, while the rats were administered metformin and HQS extract in the Met and the HQS group, respectively. All groups received continuous oral treatment for 4 weeks.

#### 4.4.2. Measurement of Biochemical Indicators

After the administration period, the rats were subjected to a 12 h fasting period, followed by blood collection from the abdominal aorta under anesthesia. The blood collected underwent centrifugation at 3000 rpm for 10 min to isolate the serum. The serum was split into two parts: one designated for the analysis of blood components and the other aimed at evaluating biochemical markers. Measurements of TG, INS, T-CHO, GHb, HDL-C, LDL-C, and KB were conducted following the manufacturer’s protocols.

#### 4.4.3. UPLC-Q/TOF-MS/MS Analysis

The analysis of metabolomics was performed utilizing UPLC-Q-TOF/MS with ESI operating in negative ion mode, and the data were gathered using Model, Control, and HQS in that order. Leucine enkephalin in combination with sodium formate served as the lock mass. The sample cone voltage was set at 40 volts, which plays a crucial role in ionization efficiency. Additionally, the source temperature was maintained at a steady 100 degrees Celsius to promote consistent sample volatilization. The capillary voltage was configured to 2.5 kilovolts, enabling effective ion extraction from the sample. Similarly, an extraction cone voltage of 40 volts was employed to assist in the separation of ions post ionization. To further refine the analysis, desolvation gas settings were optimized, with a desolvation temperature of 450 degrees Celsius and a flow rate of 800 L per hour. Finally, a cone gas flow rate of 50 L per hour was also established, contributing to the overall efficacy of the optimization process. The collision energy was established at a low level of 6 V, whereas during detection in MSE scan mode, it increased from 30 to 70 V. The process of data collection covered a mass-to-charge ratio range from 50 to 1200, with a scanning period of 0.2 s.

Considering the potential variations in retention time and elution order during UPLC-Q-TOF/MS analysis, it is important to monitor the system performance continuously. In the course of this study, we developed a QC sample by amalgamating 50 µL of all serum samples. This QC sample was utilized as a benchmark for the validation of our methodological approach. By establishing this standard, we aimed to ensure the reliability and accuracy of our testing procedures. The QC sample was analyzed three times before the sample series commenced. Subsequently, an injection was performed after every fourth sample throughout the entire analysis process. To guarantee the system’s reliability and consistency throughout the analysis, this strategy was put into practice.

#### 4.4.4. Spectrum–Effect Correlation Analysis

As an initial measure in the correlation of spectrum effects, the conditions for UPLC-Q-TOF/MS were consistent with those outlined earlier. The measurements for INS, TG, LDL-C, HDL-C, T-CHO, GHb, and KB values were performed in conformity to the protocols. Furthermore, the data processing system (DPS V15.10) was utilized to perform gray relational analysis (GRA) in order to evaluate the connection between the compounds and the serum indicators. Before carrying out the GRA, the intensities of the peak responses were normalized. The normalized data acted as the secondary sequences, while the serum marker levels were categorized as the primary sequence [31]. The correlation between the ingredients and their therapeutic effectiveness represents the mean correlation of the ingredients with various serum markers. The specific GRA procedure is as follows:

Firstly, the raw data of associated factors are normalized and then the deviation sequences are determined. Finally, the gray relational coefficients are calculated by the following equations:

Set *x_i_* = (*x_i_*(1), *x_i_*(2),…, *x_i_*(n)) as the sequence of associated factors. The correlation coefficient is defined as the following equation:$$\gamma (x_{0}(k),x_{i}(k))=\frac{\underset{i}{\min}\;\underset{k}{\min}|x_{0}(k)-x_{i}(k)|+\xi \;\underset{i}{\max}\;\underset{k}{\max}|x_{0}(k)-x_{i}(k)|}{|x_{0}(k)-x_{i}(k)|+\xi \underset{i}{\max}\;\underset{k}{\max}|x_{0}(k)-x_{i}(k)|}$$
where *ξ* is the distinctive coefficient lying between 0 and 1, which is set as 0.1.

The gray relation grade (GRG) is formulated as follows:$$\gamma (x_{0},x_{i})=\frac{1}{n}\sum_{k=1}^{n}\gamma (x_{0}(k),x_{i}(k))$$
where *n* is the number of performance characteristics.

The influence degree of the factors, including the response of components and several serum indicators on the research object, was estimated by comparing their GRG value. The higher GRG value between the two associated factors is, the closer sequence of the two factors would be.

### 4.5. Determination of 8 Active Components in HQS Extract

#### 4.5.1. Preparation of Standard and Sample Solutions

Preparation of standards: Eight reference standards were dissolved in HPLC-grade methanol to prepare stock solutions with a concentration of 1.0 mg/mL. The above solutions were then diluted to afford a series of standard working solutions that were used for quantitative analyses. All prepared solutions were stored at 4 °C prior for later use.

Preparation of samples: The HQS extract samples were ground into fine powder and well mixed. Approximately 25 mg samples were accurately weighed and dissolved in methanol (5 mL). The obtained dispersion was centrifuged at 12,000 rpm for 10 min, and the supernatant was filtered through a 0.22 μm filter membrane for ultra-high-performance liquid chromatography (UPLC) analysis. All prepared solutions were stored at 4 °C prior to analysis.

#### 4.5.2. Quantitative Analysis

Quantitative analysis was performed on a Waters ACQUITY UPLC system (Waters Corporation, Milford, MA, USA). Separations were accomplished at a flow rate of 0.2 mL/min using an ACQUITY UPLC HSS T3 column (100 mm × 2.1 mm, 1.8 μm). The mobile phase comprised solvents A (acetonitrile) and B (water), and the following gradients were used: 0–10 min, 10–15% A; 10–15 min, 15–20% A; 15–20 min, 20–30% A; 20–25 min, 30–38% A; 25–30 min, 38–39% A; 30–35 min, 100% A. The UV-detecting wavelength was 203 nm. The injection volume equaled 2 μL, and the column temperature was set to 35 °C.

### 4.6. Pharmacological Study of Active Constituent of HQS in the Treatment of T2DM

#### 4.6.1. Cell Culture

Cells, specifically HepG2, which originate from human hepatocellular carcinoma, were sourced from the Cell Resource Center located in Beijing, China. In the laboratory, these cells were cultured in high-glucose DMEM medium sourced from Gibco in the United States. To promote optimal growth conditions, the medium was supplemented with 10% fetal bovine serum, also from Gibco, (Carlsbad, CA, USA), alongside a mixture of antibiotics comprising 0.1 mg/mL penicillin and streptomycin, provided by New Cell & Molecular Biotech Co., Ltd., (Wuxi, China). The incubation conditions for the cultured cells were strictly controlled, maintaining a temperature of 37 °C and a carbon dioxide level of 5%. Additionally, to ensure the health and proliferation of the cells, refresh the culture medium three times per week.

#### 4.6.2. Cell Viability

The cell viability of HepG2 cells was assessed by CCK8 (Bio-Channel, Nanjing, China). Beginning by seeding cells at a density of 2 × 10^4^ cells per well within a 96-well microplate, this was followed by a 24 h incubation period. Following the incubation period, the cells were subjected to a treatment regimen involving a wide spectrum of concentrations for the HQS extract. Specifically, these included concentrations of 5, 10, 15, 20, 25, 30, 35, and 40 μg/mL. In addition, eight active components were evaluated at varying concentrations of 2, 4, 8, 16, 32, 64, 128, and 256 μM. The active constituent (AC, composed of 8 compounds mixed in equal proportions) was also tested at similar concentrations of 5, 10, 15, 20, 25, 30, 35, and 40 μg/mL. The solvents used for preparing the drugs are all serum-free culture media. This treatment phase was conducted for a duration of 24 h in an incubator set to maintain a 5% CO2 atmosphere at a temperature of 37 degrees Celsius. It is worth noting that a control group was maintained under the same conditions to ensure the validity of the experimental results. Each group consisted of 6 replicate wells to provide accurate and reliable data for analysis. Following treatment, introduce 10 μL of CCK8 detection solution into each well. Subsequently, incubate the samples at 37 °C for two hours and determine absorbance at 450 nm using a multifunctional microplate reader (Thermo Fisher Scientific, Waltham, MA, USA).

#### 4.6.3. Glucose Consumption Assay

The glucose oxidase–peroxidase method was utilized to assess glucose consumption. Cells were placed at a density of 2 × 10^4^ cells per well in the 96-well plate, reserving 6 wells as blanks. After treatment for 24 h, the culture medium was analyzed and the glucose concentration in the supernatant was quantified. To precisely quantify glucose consumption throughout the experiment, the glucose concentration observed in the blank was deducted from that measured in the experimental well.

#### 4.6.4. Glycogen Assay

HepG2 cells were treated with 0.25% trypsin-EDTA (Gibco, USA) to create a single-cell suspension. After this step, the cells underwent centrifugation at 1000 rpm for 10 min, during which the supernatant was discarded. Next, the sonication of the cells took place in an ice water bath. The alkaline solution to the sample was added without prior centrifugation, and the mixture was heated in boiling water for 20 min to prepare the sugar detection solution. The glycogen content was then determined following the instructions.

#### 4.6.5. Animal Models

Male C57BL/6 mice, aged 8 weeks and weighing approximately 20 ± 2 g, were obtained from the Beijing Weitong Lihua Laboratory Animal Technology Co., Ltd. (Beijing, China) (License No.: SCXK (Zhe) 2022-0002, China). These mice were housed in the Experimental Animal Center of China Pharmaceutical University, where the temperature was maintained at approximately 23 ± 2 °C and the relative humidity was set at 45 ± 10%. They were subjected to a light–dark cycle lasting 12 h and were given free access to water as well as rodent food. All experimental procedures adhered to the standards set by the Laboratory Animal Ethics and Management Committee, showcasing dedication to compliance with applicable international, national, and institutional animal welfare regulations. Approval for the study was granted by the Animal Ethics Committee of China Pharmaceutical University. This commitment to ethical standards is reflected in the study’s identification number, which is 2022-03-024.

A total of 42 C57BL/6 mice were randomly assigned to seven groups (n = 6), including the control group (Control), the model group (Model), the group treated with HQS extract (1.7 g/kg/day), the positive control group (Met, 100 mg/kg/day), and three groups receiving varying doses of a combined mixture of eight active ingredients from HQS: AC_L (50 mg/kg/day), AC_M (100 mg/kg/day), and AC_H (200 mg/kg/day), with the solvent being water. The control group was administered a normal diet, whereas the experimental groups received a high-fat and high-sugar diet to expedite the onset of diabetes. The feeding period lasted for four weeks, followed by a 12 h fasting interval. The mice in both the model group and the treatment group received i.p. injections over a period of three days. The injected substances consisted of 40 milligrams per kilogram of STZ, which was dissolved in a sodium citrate buffer solution (0.1 M, pH 4.4) and maintained in an ice bath during preparation. Conversely, the control group mice were administered an equal volume of the buffer alone. Mice showing elevated blood glucose levels (FBG ≥ 11.1 mmol/L) were classified as diabetic. After an eight-week gavage treatment, the mice underwent an 8 h fasting period prior to blood samples being collected from the orbital vein. Following this, the pancreases and livers of the mice were harvested for subsequent analysis.

#### 4.6.6. Oral Glucose Tolerance Test (OGTT) and Insulin Tolerance Test (ITT)

The consumption of food and water was tracked daily, while their FBG levels were evaluated weekly using ACCU-CHEK Active Blood Glucose Test Strips. At the close of the 8th week, an OGTT was conducted. After the treatment period, the mice were subjected to a 12 h fasting period in preparation for the OGTT procedures. The next morning, an oral glucose (2 mg/kg) was given to the mice. Obtaining the blood samples from their tail veins at predetermined intervals: 0, 30, 60, 90, 120, 150, and 180 min. The collected samples underwent analysis to determine the blood glucose, facilitating the construction of a curve for the OGTT. Furthermore, the AUC was computed to quantify the glucose tolerance response of the mice throughout the specified time period.

Conduct the ITT to assess IR of the mice. This assessment followed a fasting period of 12 h. Blood samples were collected by puncturing their tails, and the blood glucose were evaluated using an ACCU-CHEK-active glucose meter (F. Hoffman La Roche GmbH, Grenzach, Germany) to establish baseline glucose levels. Each mouse received a dosage of insulin set at 1 IU/kg through i.p. injection. Blood samples were collected at intervals of 0, 15, 30, 45, and 60 min to measure glucose levels. The resulting blood glucose measurements were utilized to create an ITT curve, from which the AUC was calculated.

#### 4.6.7. Analysis of Serum Biochemical Indicators

The kits utilized for measuring TG, T-CHO, LDL-C, and HDL-C were sourced from the Nanjing Jiancheng Bioengineering Institute, located in Nanjing, China. Additionally, the ELISA kits for determining GHb and KB levels were obtained from Jiangsu Meimian Industrial Co., Ltd., also based in Rugao, China. Each step of the experimental procedure was performed meticulously, adhering closely to the guidelines provided by the manufacturers.

#### 4.6.8. Determination of Insulin Sensitivity

Blood samples obtained from the tail vein of mice were assessed using a glucose meter. The INS levels in the mice were measured with an ELISA kit from Jiangsu Meimian Industrial Co., Ltd. (Rugao, China), meticulously adhering to the guidelines. The formula used to calculate IR via the steady-state model, known as HOMA-IR, is expressed as follows: HOMA-IR = [FBG (mmol/L) × fasting plasma insulin (mIU/L)]/22.5. This equation combines plasma INS level measurements and fasting blood glucose to estimate IR in the test subjects. On the other hand, the formula for assessing INS sensitivity through the homeostasis model (HOMA-β) is as follows: HOMA-β = 20 × fasting plasma insulin (mIU/L)/FBG (mmol/L) - 3.5. This equation enables the assessment of INS sensitivity by connecting fasting INS concentrations to FBG, providing insights into the metabolic condition.

#### 4.6.9. Histological Examination

During the collection of the tissue samples, small fragments of mouse pancreatic and liver tissue were preserved in a 4% paraformaldehyde solution for a duration of 24 h. This preservation process was followed by the embedding of the samples in paraffin and subsequent sectioning [32]. Histological alterations in the liver were assessed using H&E staining as well as PAS staining techniques, both of which were observed under a microscope at a magnification of 400×. In contrast, the pancreatic tissue changes were evaluated through H&E staining, with observations made at a lower magnification of 100×.

### 4.7. Metabolomics Analysis

The processes for acquiring plasma samples and conducting UPLC-Q/TOF-MS/MS analyses aligned with those used in spectrum–effect correlation assessments. The subjects were categorized into three groups: the control group, the model group, and the AC group (100 mg/kg/day). The raw data from UPLC-Q-TOF/MS were processed utilizing the XCMS software (V3)within the R programming environment. Subsequently, PCA, PLS-DA, and OPLS-DA analyses were carried out using MetaboAnalyst 5.0 (http://www.metaboanalyst.ca/, accessed on 2 February 2023), a thorough online tool for analyzing, visualizing, and interpreting metabolomic data. Principal Component Analysis (PCA) was employed to evaluate the metabolic profiles across all groups. To establish a model that correlates metabolite expression with sample classifications, Partial Least Squares Discriminant Analysis (PLS-DA) was utilized. Furthermore, Orthogonal Partial Least Squares Discriminant Analysis (OPLS-DA) was applied to minimize the variation among different groups and to identify metabolites that are differentially expressed in relation to Type 2 diabetes mellitus (T2DM). The quality of the model was assessed using model fitness metrics (R^2^) and predictive capability (Q^2^), with higher R^2^ values (approaching 1) and Q^2^ values (intersecting the negative half of the axis) indicating a well-performing model.

The potential biomarkers were meticulously screened using specific criteria that included a *p*-value ≤ 0.05 and a VIP ≥ 1.0. After establishing this selection process, the structures of the prospective biomarkers were systematically identified through various reputable databases, which included the HMDB accessible at http://www.hmdb.ca/, accessed on 8 February 2023, the PubChem database available at https://pubchem.ncbi.nlm.nih.gov/, accessed on 8 February 2023, as well as the METLIN database found at https://metlin.scripps.edu/, accessed on 8 February 2023, among other complementary resources. Once the biomarkers were identified, they were imported into the MetaboAnalyst 5.0 database, reachable at https://www.metaboanalyst.ca/, accessed on 9 February 2023. This facilitated a comprehensive analysis of metabolic pathways, employing the KEGG database, which can be accessed at http://www.genome.jp/kegg/, accessed on 9 February 2023. Through this analysis, significant metabolic pathways relevant to the studied biomarkers were revealed, enhancing our understanding of their biological implications and roles within metabolic processes.

### 4.8. Molecular Docking

Kaempferol, Tiliroside, Scopoletin, Moracin A, Astragaloside IV, Astragaloside III, Formononetin, and Puerarin were identified as potential active constituents of HQS in the management of T2DM. Through metabolomics, the primary target proteins associated with HQS in type 2 diabetes were evaluated. To obtain CAS numbers for the small molecules, the PubChem database (https://PubChem.ncbi.nlm.nih.gov/, accessed on 15 February 2023) was searched, followed by downloading the compounds in SDF format. Open Babel 3.1.1 was utilized to convert these files into MOL2 format. The essential protein 3D structures were obtained from the PDB database (http://www.rcsb.org/PDB, accessed on 15 February 2023) by filtering based on species, resolution of conformations, sequence fidelity, and pH levels, and saved as PDB files for docking processes. The receptor was prepared using Autodocktools 1.5.7 after dehydrogenating and hydrogenating, while the ligands were treated similarly. The interactions between active compounds and core target proteins were optimized. Following the execution of AutoGrid and AutoDock, the docking results of the compounds with the proteins were acquired. Consequently, a visualized heat map was generated using PyMOL 2.5 software.

### 4.9. Statistical Analysis

Statistical evaluations were performed using SPSS software version 22.0 (IBM Corp., Armonk, NY, USA). Based on the evaluations of normality and variance homogeneity, the data analysis utilized either the one-way ANOVA or the rank sum test, considering a *p*-value of less than 0.05 as statistically significant.

## 5. Conclusions

This research employed network pharmacology and spectrum–effect correlation analysis to obtain the active constituent of HQS for the intervention of T2DM. Both in vivo and in vitro experiments were conducted to evaluate the effects of AC. Non-targeted metabolomics and molecular docking were utilized to explore potential therapeutic pathways. This study supports the scientific basis for prioritizing AC in the assessment of TCM, provides a foundation for the clinical application of HQS, and demonstrates new avenues for TCM drug development. The current results indicate a trend towards a superior efficacy of AC, although this trend lacks statistical significance. Therefore, future research should concentrate on optimizing the proportions of the components in AC and clarify the synergistic effect among each component to better emphasize its superior efficacy. Furthermore, this study has limitations: only one cell line was utilized in vitro, which may limit the generalizability of the findings. The mechanism by which AC treats T2DM was not further validated, and the in vivo processes of the eight active ingredients remain inadequately defined. Future research will focus on addressing these issues.

## Figures and Tables

**Figure 1 ijms-26-04578-f001:**
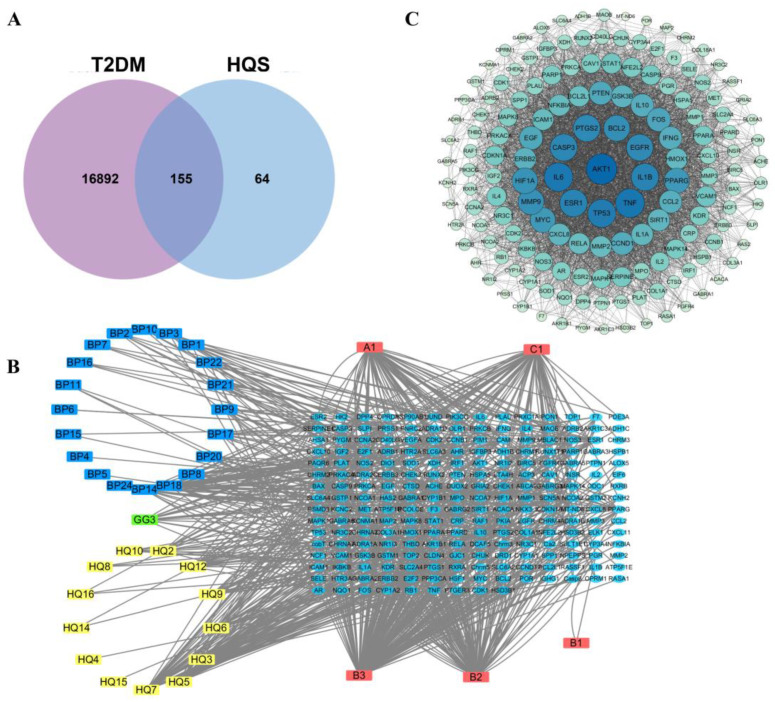
(**A**) The common targets of HQS together with the differentially expressed genes related to T2DM; (**B**) the network of compounds and targets; (**C**) the PPI network encompassing genes associated with the treatment of T2DM using HQS.

**Figure 2 ijms-26-04578-f002:**
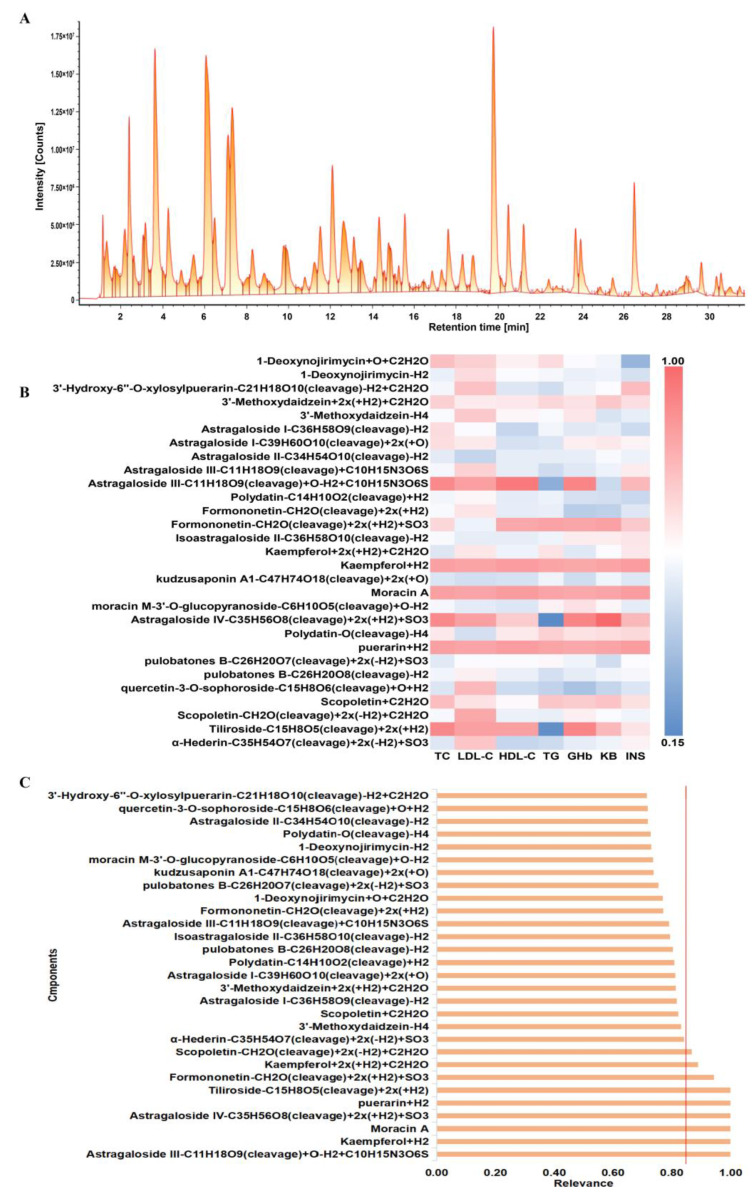
Outcomes of spectrum–effect correlation analysis. (**A**) A total ion chromatogram showcasing blood components extracted from HQS in rat models; (**B**) a heat map illustrating correlations between HQS blood components and serum biomarkers; (**C**) the relationship between HQS components and the overall evaluation metric, with the red line representing a relevancy value of 0.85.

**Figure 3 ijms-26-04578-f003:**
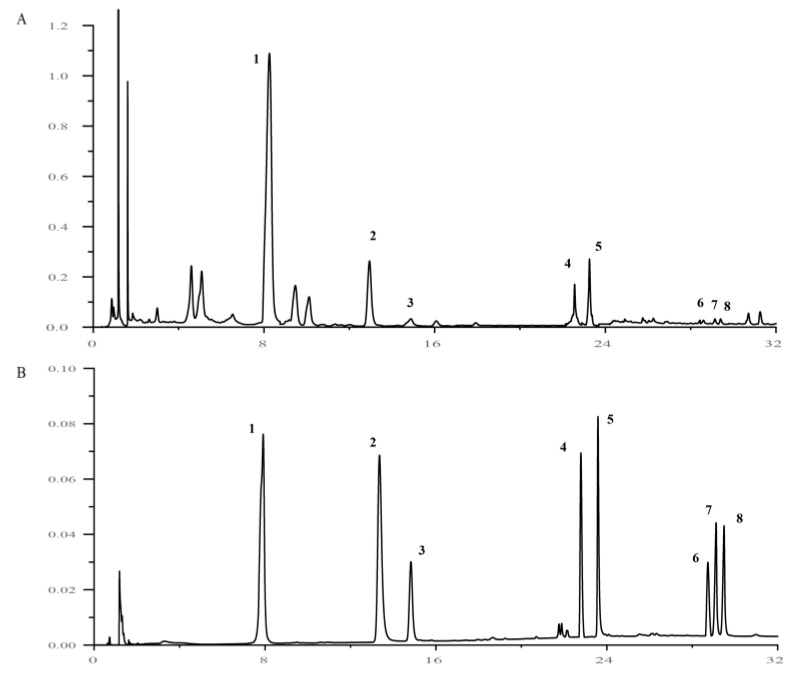
(**A**) Chromatogram of HQS extract. (**B**) Chromatogram of the standard substance; 1–8 are, respectively, Puerarin, Scopoletin, Moracin A, Tiliroside, Kaempferol, Formononetin, Astragaloside IV, and Astragaloside III.

**Figure 4 ijms-26-04578-f004:**
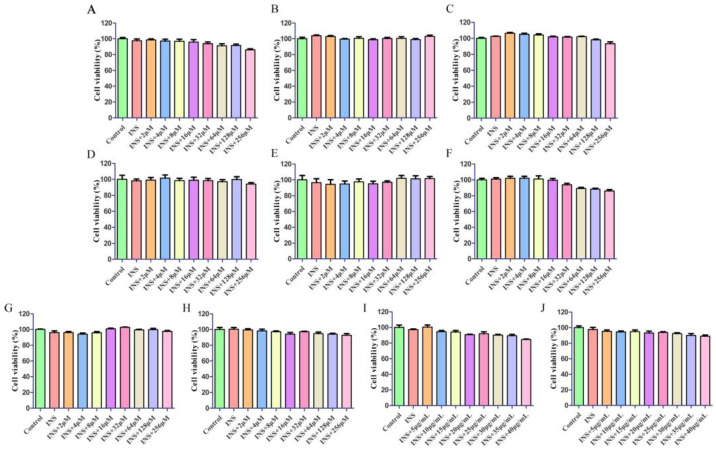
Cell viability of IR-HepG2 cells induced by INS. Astragaloside IV (**A**), Astragaloside III (**B**), Kaempferol (**C**), Tiliroside (**D**), Formononetin (**E**), Scopoletin (**F**), Moracin A (**G**), Puerarin (**H**), HQS extract (**I**), and AC (**J**).

**Figure 5 ijms-26-04578-f005:**
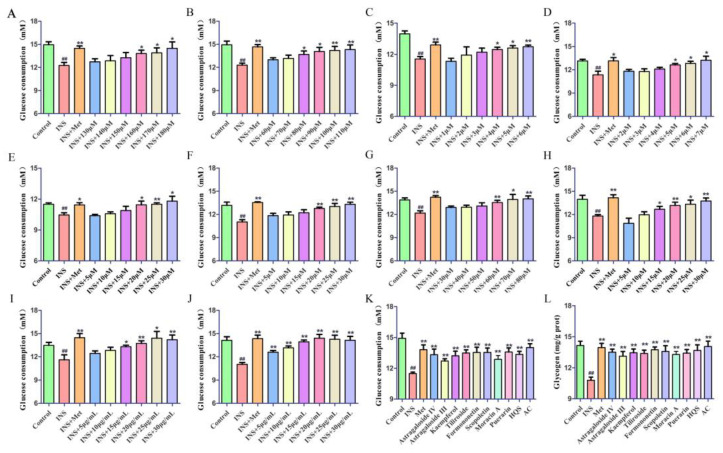
Beneficial effect curve in IR-HepG2 cells. Astragaloside IV (**A**), Astragaloside III (**B**), Kaempferol (**C**), Tiliroside (**D**), Formononetin (**E**), Scopoletin (**F**), Moracin A (**G**), Puerarin (**H**), HQS extract (**I**), and AC (**J**); effect of HQS on glucose consumption (**K**) and glycogen levels (**L**). In comparison to the control group, ^##^ *p* < 0.01; when compared to the model group, * *p* < 0.05, ** *p* < 0.01.

**Figure 6 ijms-26-04578-f006:**
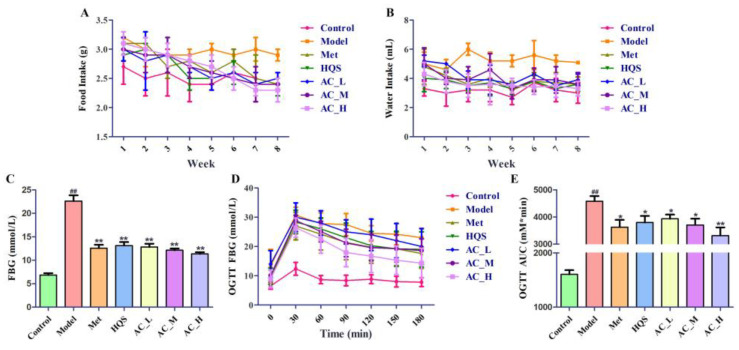
The hypoglycemic effects of the HQS extract and AC were evaluated in mice induced with T2DM using STZ. Over a duration of 8 weeks, the researchers monitored several parameters related to the mice’s health. Food intake (**A**), water intake (**B**), FBG levels (**C**), and the results of OGTT (**D**,**E**). In comparison to the control group, ^##^ *p* < 0.01; when contrasted with the model group, * *p* < 0.05, ** *p* < 0.01.

**Figure 7 ijms-26-04578-f007:**
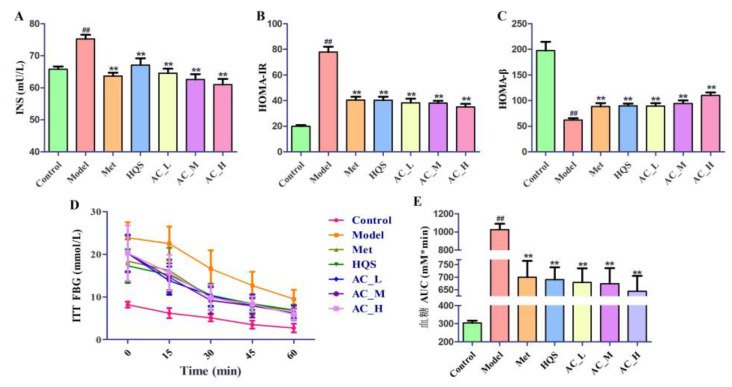
The effect of HQS on insulin sensitivity in T2DM mice. (**A**) Serum INS levels; (**B**) HOMA-IR; (**C**) HOMA-β; (**D**) curve of ITT; (**E**) AUC of ITT. In comparison to the control group, ^##^ *p* < 0.01; when contrasted with the model group, ** *p* < 0.01.

**Figure 8 ijms-26-04578-f008:**
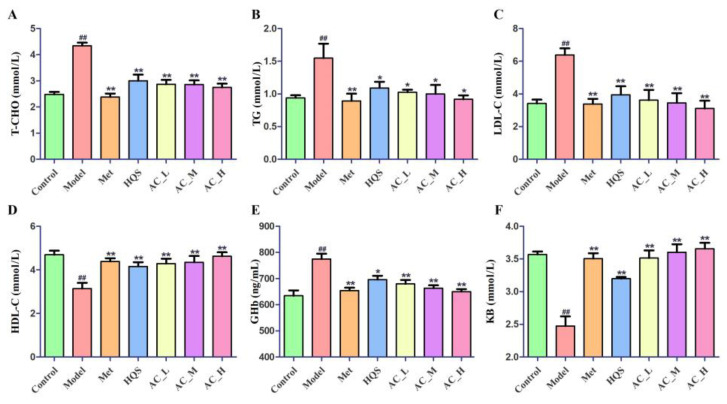
Enhancement of HQS extract and the AC on biochemical markers in mice T2DM. The levels of serum T-CHO (**A**), TG (**B**), LDL-C (**C**), HDL-C (**D**) GHb (**E**) and KB (**F**) in mice. Relative to the control group, ^##^ *p* < 0.01; when compared to the model group, * *p* < 0.05, ** *p* < 0.01.

**Figure 9 ijms-26-04578-f009:**
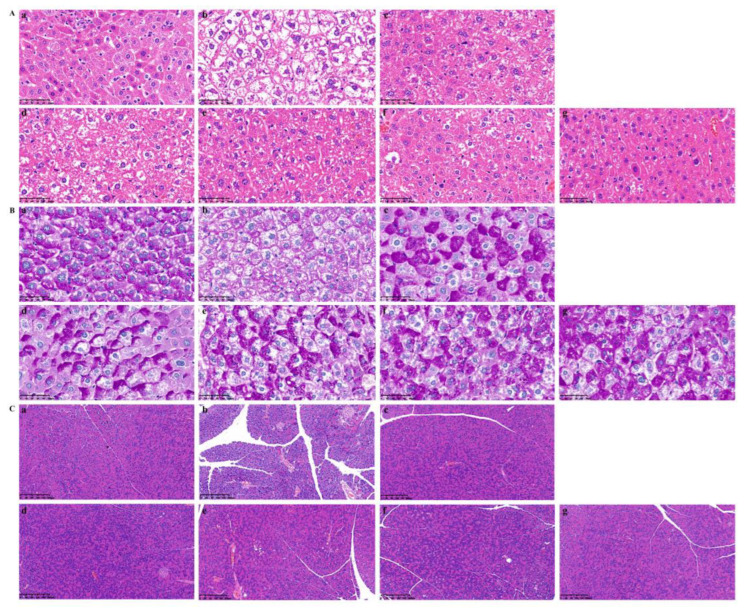
Histological evaluation of the effects of HQS extract and active constituent on liver and pancreas tissues in STZ-induced T2DM mice. In the results of H&E staining (**A**) and PAS staining (**B**) of the liver (magnification, ×400), as well as H&E staining of the pancreas (**C**) (magnification, ×100), a–g represent the control group, model group, Metformin (Met) group, HQS group, and AC groups (50, 100, 200 mg/kg).

**Figure 10 ijms-26-04578-f010:**
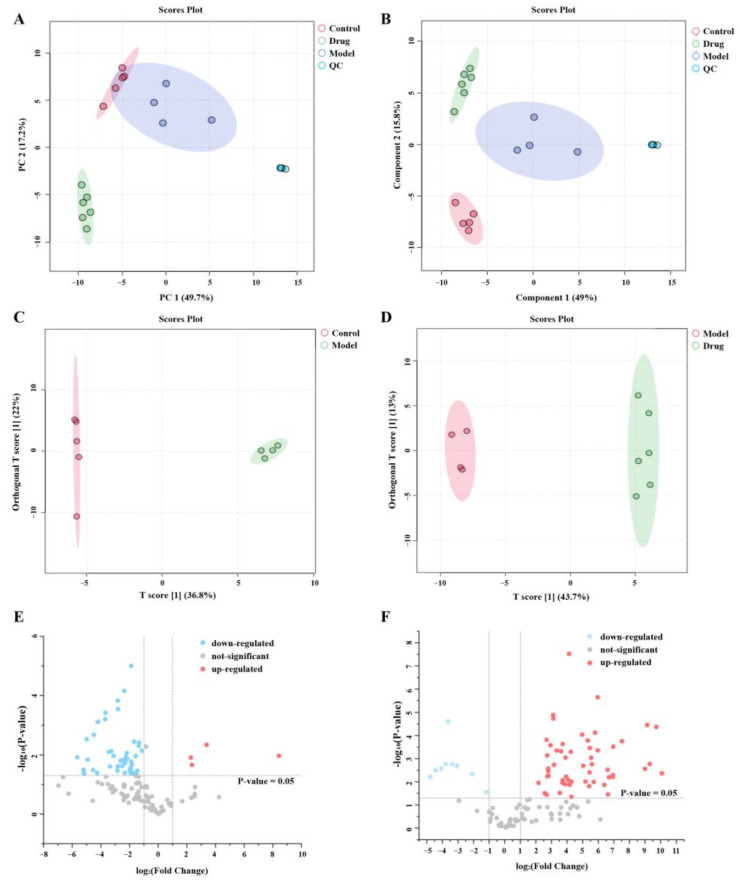
Analysis using multivariate statistics. (**A**) PCA score plots; (**B**) PLS-DA score plots. Distribution of samples across various groups in OPLS-DA mode: (**C**) Control vs. Model in negative ion mode; (**D**) Model vs. HQS in negative ion mode. Variation demonstrated in volcano plots: (**E**) Control vs. Model; (**F**) Model vs. HQS. Note: The horizontal axis represents change multiples, while the vertical axis indicates *p*-values from *t*-tests.

**Figure 11 ijms-26-04578-f011:**
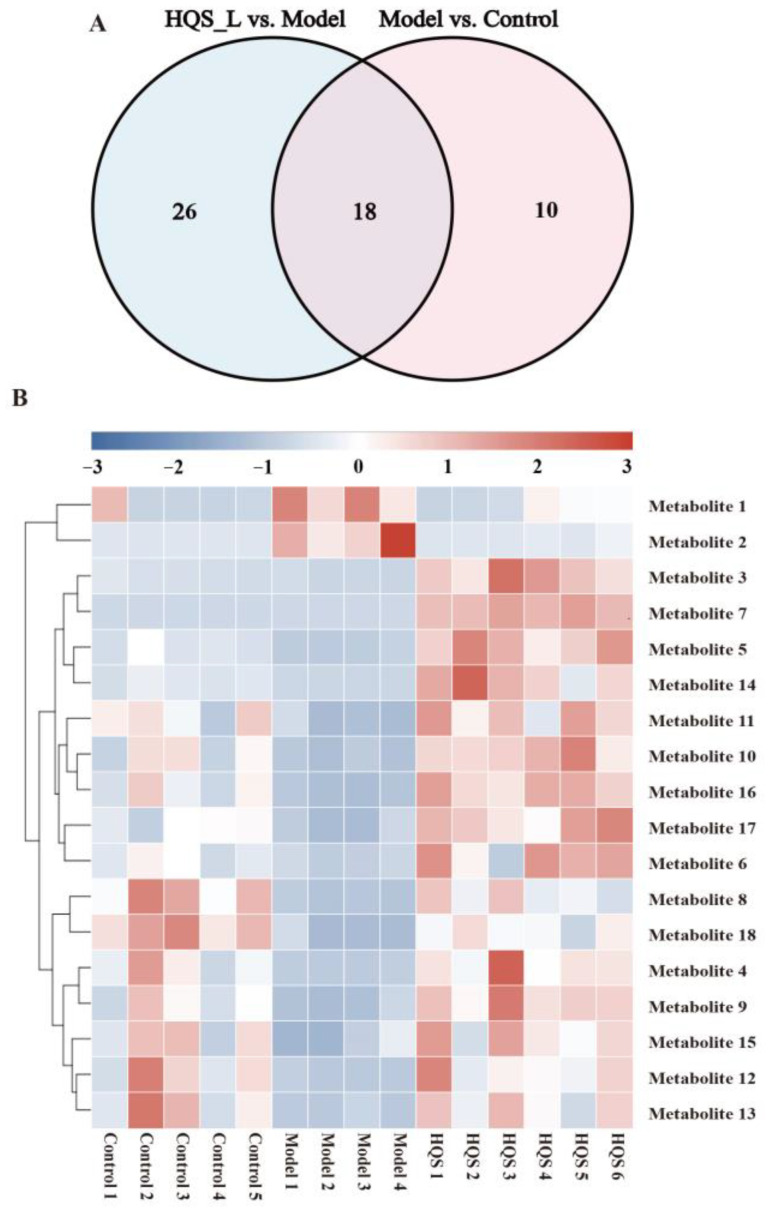
(**A**) Venn diagram showing the overlapping metabolites; (**B**) the heatmap displaying the clustering of differential metabolite levels. Red denotes an increase in metabolite levels, while blue signifies a decrease in metabolite levels.

**Figure 12 ijms-26-04578-f012:**
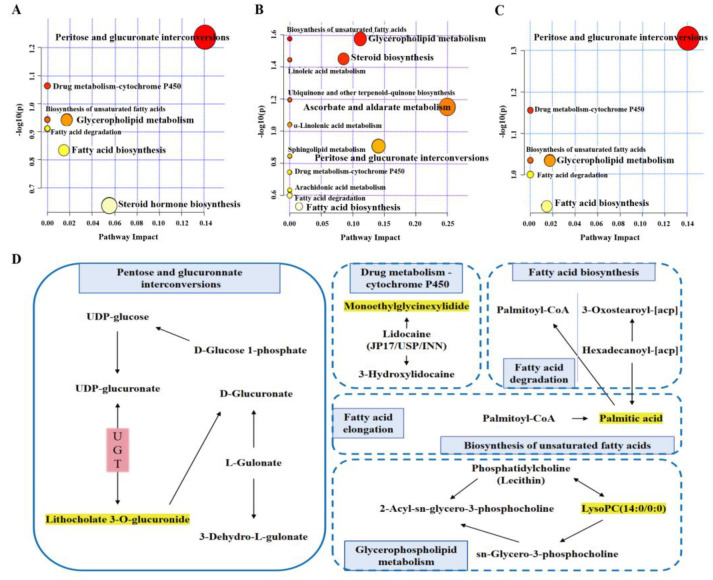
Bubble diagram illustrating the analysis of differential metabolite pathways in rat serum: (**A**) Control vs. Model; (**B**) Model vs. HQS; (**C**) comparison between (**A**) and (**B**). Each point denotes a specific metabolic pathway; the size of the dot and its color intensity are positively associated with the significance of the metabolic pathway. (**D**) Analysis of metabolic pathways. The metabolites highlighted in yellow are the potential biomarkers identified in this study.

**Figure 13 ijms-26-04578-f013:**
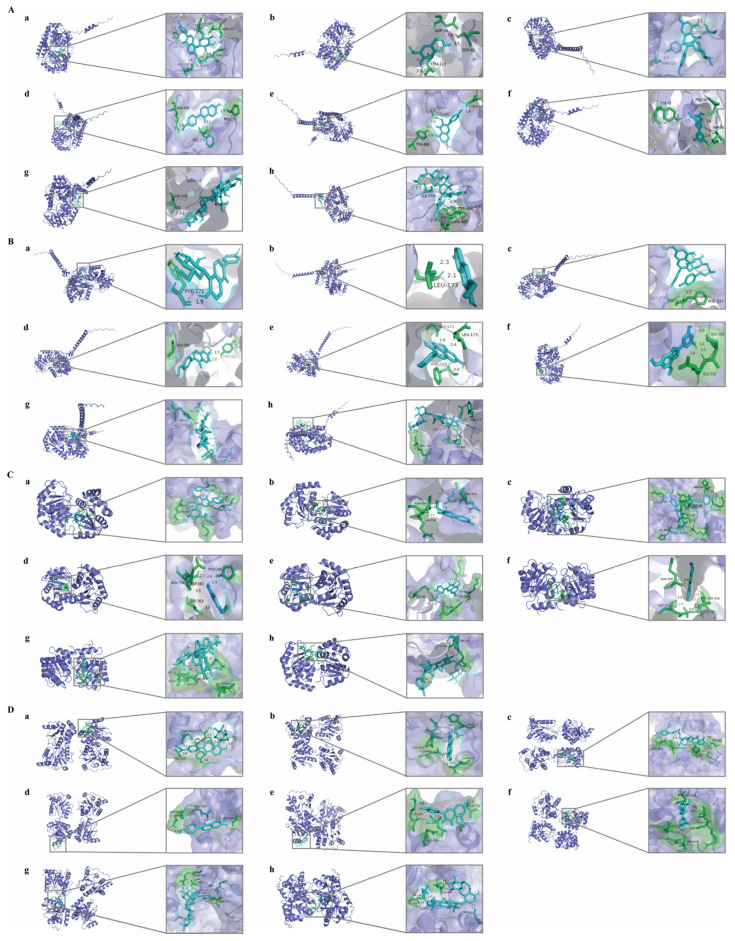
Molecular docking of 8 representative compounds of HQS with UGT1A1 (**A**), UGT1A9 (**B**), UGT2B10 (**C**), and UGT2B15 (**D**). (a) Puerarin, (b) Scopoletin, (c) Tiliroside, (d) Formononetin, (e) Kaempferol, (f) Moracin A, (g) Astragaloside IV, and (h) Astragaloside III. The compounds under investigation are represented in cyan, while the adjacent residues within the binding pockets are shown in green. Additionally, the receptor’s backbone is illustrated as a slate-colored cartoon.

**Table 1 ijms-26-04578-t001:** Characteristic parameters of the 28 active components of HQS in the network.

No.	ID	Components	Degree	Betweenness	Closeness
1	B3	Quercetin	161	0.63	0.56
2	B2	Kaempferol	113	0.14	0.42
3	A1	Formononetin	69	0.10	0.39
4	C1	Beta-sitosterol	68	0.13	0.39
5	HQ7	7-O-Methylisomucronulatol	40	0.08	0.39
6	HQ5	Isorhamnetin	33	0.06	0.38
7	BP14	Iristectorigenin	30	0.01	0.37
8	BP18	Tiliroside	24	0.01	0.36
9	HQ3	α-Hederin	22	0.05	0.37
10	HQ6	3′-Hydroxy-6″-O-xylosylpuerarin	21	0.02	0.37
11	HQ12	Pulobatones B	19	0.01	0.37
12	HQ9	Astragaloside III	19	0.01	0.36
13	GG3	Puerarin	18	0.01	0.36
14	BP8	Sanggenone F	15	0.01	0.36
15	BP20	Kudzusaponin A_1_	13	0.02	0.36
16	BP17	Moracin A	12	0.00	0.36
17	HQ2	Astragaloside IV	12	0.00	0.36
18	BP9	Sanggenone H	10	0.01	0.35
19	BP1	6″-O-Acetyldaidzin	8	0.01	0.35
20	BP21	Quercetin 3-O-(6″-galloyl)-β-D-glucopyranoside	8	0.00	0.35
21	BP22	Scopoletin	8	0.00	0.35
22	BP3	M-3′-O-glucopyranoside	6	0.00	0.35
23	BP10	Mulberroside C	5	0.00	0.35
24	BP2	Mairin	5	0.00	0.25
25	BP16	Moracin C	4	0.00	0.32
26	HQ11	3′-Methoxydaidzein	3	0.00	0.34
27	HQ16	Corchoionoside C	3	0.00	0.35
28	B1	Mairin	2	0.00	0.25

**Table 2 ijms-26-04578-t002:** MS data of 57 blood components (peaks) from rats.

No.	Identified Cmponents	M (*m*/*z*)	Formular	Error (ppm)	Adduct
1	Astragaloside III-C_11_H_18_O_9_(cleavage) + O-H_2_ + C_10_H_15_N_3_O_6_S	808.401	C_40_H_63_N_3_O_12_S	−5.5	-H
2	Astragaloside IV-C_35_H_56_O_8_(cleavage) + 2×(+H_2_) + SO_3_	263.044	C_6_H_16_O_9_S	−0.9	-H
3	Kaempferol + H_2_	287.06	C_15_H_12_O_6_	11.7	-H
4	Moracin A	285.08	C_16_H_14_O_5_	11.1	-H
5	Puerarin + H_2_	417.12	C_21_H_22_O_9_	1.7	-H
6	Tiliroside-C_15_H_8_O_5_(cleavage) + 2×(+H_2_)	375.13	C_15_H_22_O_8_	1.2	+HCOO
7	Moracin M-3′-O-glucopyranoside-C_6_H_10_O_5_(cleavage) + O	257.049	C_14_H_10_O_5_	13.6	-H
8	Formononetin-CH_2_O(cleavage) + 2×(+H_2_) + SO_3_	321.044	C_15_H_14_O_6_S	−1	-H
9	Kaempferol + 2×(+H_2_) + C_2_H_2_O	377.086	C_17_H_16_O_7_	−3.9	+HCOO
10	Scopoletin-CH_2_O(cleavage) + 2×(-H_2_) + C_2_H_2_O	199.007	C_11_H_4_O_4_	17.2	-H
11	α-Hederin-C_35_H_54_O_7_(cleavage) + 2×(-H_2_) + SO_3_	222.992	C_6_H_8_O_7_S	−0.7	-H
12	3′-Methoxydaidzein-H_4_	279.032	C_16_H_8_O_5_	6.8	-H
13	Scopoletin + C_2_H_2_O	279.055	C_12_H_10_O_5_	14.9	+HCOO
14	Astragaloside I-C_36_H_58_O_9_(cleavage)-H_2_	277.06	C_9_H_12_O_7_	13.1	+HCOO
15	3′-Methoxydaidzein + 2×(+H_2_) + C_2_H_2_O	329.107	C_18_H_18_O_6_	11.5	-H
16	Astragaloside I-C_39_H_60_O_10_(cleavage) + 2×(+O)	257.048	C_6_H_12_O_8_	−11.9	+HCOO
17	Polydatin-C_14_H_10_O_2_(cleavage) + H_2_	181.071	C_6_H_14_O_6_	−6.1	-H
18	Pulobatones B-C_26_H_20_O_8_(cleavage)-H_2_	161.046	C_6_H_10_O_5_	0.9	-H
19	Isoastragaloside II-C_36_H_58_O_10_(cleavage)-H_2_	173.045	C_7_H_10_O_5_	−4.2	-H
20	Astragaloside III-C_11_H_18_O_9_(cleavage) + C_10_H_15_N_3_O_6_S	794.426	C_40_H_65_N_3_O_11_S	−1.2	-H
21	Formononetin-CH_2_O(cleavage) + 2×(+H_2_)	241.082	C_15_H_14_O_3_	−20	-H
22	1-Deoxynojirimycin + O+C_2_H_2_O	266.088	C_8_H_15_NO_6_	−0.4	+HCOO
23	Pulobatones B-C_26_H_20_O_7_(cleavage) + 2×(-H_2_) + SO_3_	254.983	C_6_H_8_O_9_S	4.1	-H
24	Kudzusaponin A_1_-C_47_H_74_O_18_(cleavage) + 2×(+O)	243.033	C_5_H_10_O_8_	−12.4	+HCOO
25	1-Deoxynojirimycin-H_2_	206.067	C_6_H_11_NO_4_	0.5	+HCOO
26	Polydatin-O(cleavage)-H_4_	415.104	C_20_H_18_O_7_	0.3	+HCOO
27	Astragaloside II-C_34_H_54_O_10_(cleavage)-H_2_	173.045	C_7_H_10_O_5_	−4.1	-H
28	Quercetin-3-O-sophoroside-C_15_H_8_O_6_(cleavage) + O + H_2_	359.119	C_12_H_24_O_12_	−0.8	-H
29	3′-Hydroxy-6″-O-xylosylpuerarin-C_21_H_18_O_10_(cleavage)-H_2_ + C_2_H_2_O	173.045	C_7_H_10_O_5_	−5.5	-H
30	Moracin A + H_2_	333.094	C_16_H_16_O_5_	−12.1	+HCOO
31	Mulberroside C + O + H_2_	475.161	C_24_H_28_O_10_	−0.9	-H
32	Moracin B-CH_2_(cleavage) + H_2_ + C_2_H_2_O	315.09	C_17_H_16_O_6_	8.1	-H
33	Scopoletin-H_4_	187.006	C_10_H_4_O_4_	14.9	-H
34	Pulobatones B-C_26_H_20_O_8_(cleavage) + 2×(+H_2_) + SO_3_	293.055	C_6_H_16_O_8_S	−0.7	+HCOO
35	Puerarin	415.104	C_21_H_20_O_9_	0.3	-H
36	Moracin A-CH_2_(cleavage)	271.064	C_15_H_12_O_5_	11.4	-H
37	Astragaloside I-C_39_H_60_O_10_(cleavage)	179.055	C_6_H_12_O_6_	−6.1	-H, +HCOO
38	Astragaloside I-C_36_H_58_O_9_(cleavage) + 2×(+H_2_)	283.104	C_9_H_18_O_7_	0.7	+HCOO
39	Scopoletin-CH_2_(cleavage) + O+SO_3_	272.971	C_9_H_6_O_8_S	0.2	-H
40	Corchoionoside C-C_6_H_10_O_5_(cleavage)-H_2_ + SO_3_	301.076	C_13_H_18_O_6_S	3.3	-H
41	Kudzusaponin A_1_-C_47_H_74_O_18_(cleavage) + O+H_2_	229.053	C_5_H_12_O_7_	−15.5	+HCOO
42	Polydatin-C_6_H_10_O_5_(cleavage) + O	243.062	C_14_H_12_O_4_	−16.3	-H
43	6″-O-Acetyldaidzin-C_15_H_8_O_3_(cleavage) + H_2_	269.087	C_8_H_16_O_7_	−3.9	+HCOO
44	Scopoletin-CH_2_(cleavage) + 2×(-H_2_) + C_2_H_2_O	215.002	C_11_H_4_O_5_	18.4	-H
45	Scopoletin-O(cleavage)-H_2_ + C_2_H_2_O	215.032	C_12_H_8_O_4_	−14.3	-H
46	Scopoletin-O(cleavage)-H_4_	171.011	C_10_H_4_O_3_	13.7	-H
47	Quercetin 3-O-(6″-galloyl)-β-D-glucopyranoside-C_21_H_18_O_11_(cleavage) + SO_3_	248.973	C_7_H_6_O_8_S	8.8	-H
48	Sanggenone F + O + C_2_H_2_O	411.108	C_22_H_20_O_8_	−1.9	-H
49	Mulberroside C-C_19_H_16_O_4_(cleavage) + 2×(+H_2_) + SO_3_	279.038	C_5_H_14_O_8_S	−3.1	+HCOO
50	Tectorigenin-CH_2_O(cleavage)	269.049	C_15_H_10_O_5_	13.2	-H
51	Astragaloside IV-C_36_H_58_O_9_(cleavage)	149.045	C_5_H_10_O_5_	−4.6	-H
52	Moracin C + H_2_ + C_2_H_2_O	399.146	C_21_H_22_O_5_	2.1	+HCOO
53	3′-Methoxydaidzein-CH_2_O(cleavage) + SO_3_	333.008	C_15_H_10_O_7_S	1.1	-H
54	Tiliroside-C_15_H_8_O_6_(cleavage)-H_2_ + C_2_H_2_O	395.096	C_17_H_18_O_8_	−4.9	+HCOO
55	3′-Methoxydaidzein-CH_2_O(cleavage)	253.05	C_15_H_10_O_4_	−2.1	-H
56	Scopoletin-CH_2_O(cleavage)-H_2_ + C_2_H_2_O	201.022	C_11_H_6_O_4_	14.9	-H
57	Scopoletin-CH_2_(cleavage)-H_4_	172.991	C_9_H_2_O_4_	16.7	-H

**Table 3 ijms-26-04578-t003:** Contents of 8 analytes in HQS-effective extraction samples.

Compounds	Linear Range (μg)	Regression Equation	r^2^	Content (%)
Puerarin	0.33–2.00	y = 6,259,581x + 7,691,406	0.9992	42.75
Scopoletin	0.26–1.58	y = 50,561,849x – 1,061,227	1.0000	12.66
Tiliroside	0.10–0.60	y = 19,048,338x – 322,554	0.9998	2.16
Formononetin	0.11–0.63	y = 29,669,618x – 394,826	0.9997	0.25
Kaempferol	0.14–0.83	y = 15,037,786x – 719,796	0.9995	0.25
Moracin A	0.12–0.73	y = 18,485,874x – 308,581	1.0000	0.34
Astragaloside IV	1.25–7.51	y = 221,315x – 38,913	0.9997	0.24
Astragaloside III	1.28–7.68	y = 212,790x – 34,537	0.9997	0.28

**Table 4 ijms-26-04578-t004:** Identification results of 18 differential metabolites in plasma based on UPLC-Q-TOF/MS.

	**No.**	**tR** **(min)**	**Identification**	** *m* ** **/*z***	**Formula**	**MS/MS**
** 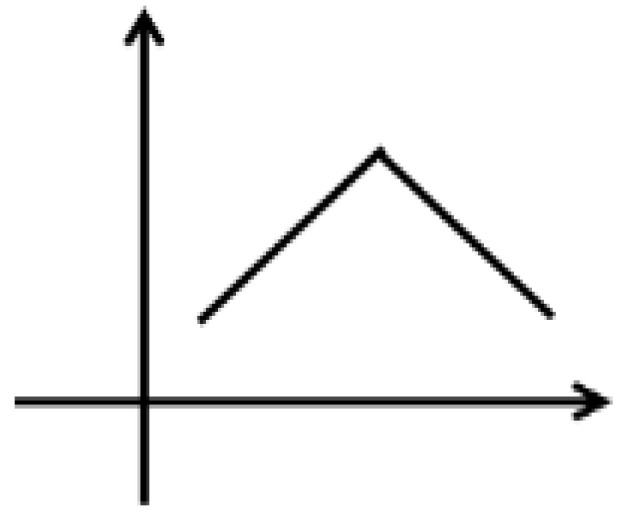 **	1	35.18	Cucurbitacin C	541.3165	C_32_H_48_O_8_	236.1204, 99.9510, 139.0171, 153.0488
	2	35.21	(3b, 16a, 20R)-3, 16, 20, 22, 25-Pentahydroxy-5-cucurbiten-11-one 3-[rhamnosyl-(1→4)-[glucosyl-(1→6)]-glucoside]	1035.5399	C_48_H_80_O_20_	293.2168, 813.4589, 975.5179
** 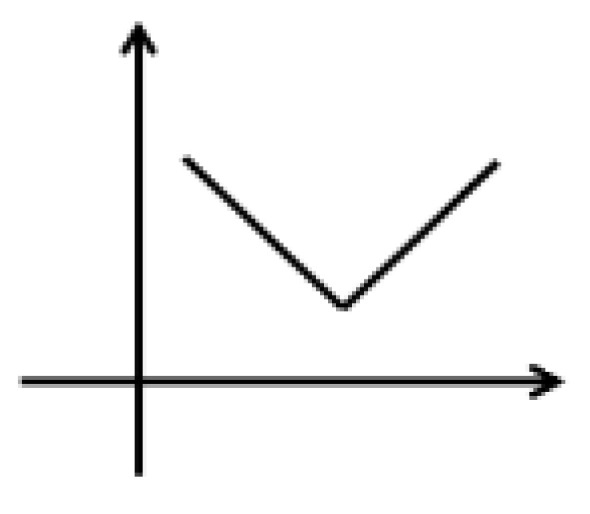 **	3	36.36	Palmitic acid	301.2389	C_16_H_32_O_2_	171.0523, 161.1419
	4	38.91	5-Aminopentanal	302.2422	C_5_H_11_NO	116.9497, 135.9911
	5	39.94	Octadecanol	305.2621	C_18_H_38_O	205.2485, 177.1300, 179.0922
	6	36.38	DG (20:5(5Z, 8Z, 11Z, 14Z, 17Z)/18:1(9Z)/0:0)	319.2491	C_41_H_68_O_5_	258.2428, 291.2491, 251.2767, 257.2859, 265.1507
	7	34.34	LysoPE (0:0/18:2(9Z, 12Z))	476.2755	C_23_H_44_NO_7_P	214.0616, 62.0303, 78.9856, 122.0292
	8	32.81	LysoPC (14:0/0:0)	502.2679	C_22_H_46_NO_7_P	220.9695, 224.1195
	9	32.81	LysoPE (0:0/16:0)	512.2963	C_21_H_44_NO_7_P	162.9853, 213.0589, 216.8936
	10	34.05	(17alpha, 23S)-17, 23-Epoxy-29-hydroxy-27-norlanosta-1, 8-diene-3, 15, 24-trione	527.3059	C_29_H_40_O_5_	243.1236, 221.1888
	11	1.32	Monoethyl glycinexylidide	243.0917	C_12_H_18_N_2_O	148.0930, 146.9884, 182.0120, 151.0939, 152.0748, 153.0817
	12	33.39	Doramapimod	586.3011	C_33_H_43_N_5_O_6_	217.8901, 218.8950, 219.8961
	13	33.40	Lithocholate 3-O-glucuronide	587.3033	C_30_H_48_O_9_	218.9011, 222.8947
	14	40.16	DG (15:0/22:5(4Z, 7Z, 10Z, 13Z, 16Z)/0:0)	609.4869	C_40_H_68_O_5_	257.2761, 256.2816, 289.0931, 251.2410, 257.2761, 265.0006
	15	33.41	Dihydrozeatin-9-N-glucoside-O-glucoside	658.2158	C_22_H_35_N_5_O_11_	202.9312, 236.1456
	16	36.39	PS (18:1(9Z)/20:4(5Z, 8Z, 11Z, 14Z))	828.5139	C_44_H_76_NO_10_P	291.2666, 321.2870, 280.2745
	17	1.43	Glutamyl methionine	277.0879	C_10_H_18_N_2_O_5_S	160.8872, 161.0873, 162.0453
	18	34.42	AzIV	1083.5434	C_54_H_84_O_22_	269.2888, 97.0081, 249.0190, 323.2561

MS/MS: ion fragmentation profile. The first column shows the trend of metabolites No. 1 and 2 increased significantly in the model group and reduced after treatment; metabolites 3–18 showed the opposite trend.

**Table 5 ijms-26-04578-t005:** Pathway analysis based on 18 potential biomarkers in serum sample.

No.	Pathway Name	Expected	Raw *p*	−log(*p*)	Impact
1	Pentose and glucuronate interconversions	0.047714	0.046913	1.3287	0.14062
2	Drug metabolism—cytochrome P450	0.071571	0.06974	1.1565	0
3	Biosynthesis of unsaturated fatty acids	0.095427	0.092155	1.0355	0
4	Glycerophospholipid metabolism	0.095427	0.092155	1.0355	0.01736
5	Fatty acid elongation	0.10338	0.099536	1.002	0
6	Fatty acid degradation	0.10338	0.099536	1.002	0
7	Fatty acid biosynthesis	0.12459	0.119	0.92446	0.01472

**Table 6 ijms-26-04578-t006:** Docking scores of different ligands binding with UGTs.

Ligand	Receptor	Docking Score (kJ/mol)	Docking Sites
Puerarin	UGT2B15	−24.23	Gly-365, His-387, Tyr-386, Ser-447, His-450
Scopoletin		−24.69	His-375, Thr-374, Gly-311, Ser-309, Asn-379, Gln-360, Leu-310
Tiliroside		−24.64	His-281, Lys-284, Tyr-382, Asn-403, Asp-402
Formononetin		−25.06	Gly-365, Leu-289, Lys-291
Kaempferol		−22.34	Ala-286, Asp-362, Leu-289, Glu-294, Lys-291
Moracin A		−25.73	Ile-314, Lys-343, Lys-344, Asn-323, Asn-346
Astragaloside IV		−19.83	Ser-315, Met-31, Ser-415, Asp-417, Arg-419
Astragaloside III		−21.71	Ser-432, Glu-319, Met-317, Ile-314, Ser-315
Puerarin	UGT2B10	−28.62	His-385, Thr-367, Met-279, Lys-282, Lys-285
Scopoletin		−24.98	Gln-358, Glu-381, Asn-359
Tiliroside		−30.33	Lys-443, Arg-446, Tyr-384, Lys-438, Met-279, Leu-362, Ala-278, His-364, Lys-285
Formononetin		−26.86	Pro-286, Asp-360, Gly-363, Asn-359
Kaempferol		−26.11	Lys-285, Lys-282, Tyr-384, Thr-367, Lys-438
Moracin A		−30.12	Asn-359, Asp-360, Gln-358
Astragaloside IV		−19.79	Asp-344, Leu-352, Lys-342, Asn-341, Gly-340, Asp-339
Astragaloside III		−25.48	Trp-355, Lys-354, Glu-290
Puerarin	UGT1A1	−19.50	His-372, Leu-355, Gln-357, His-376,Asn-279, Asn-358
Scopoletin		−19.75	Asp-36, Leu-123, Ser-65
Tiliroside		−19.66	Ser-120, Gly-395
Formononetin		−24.14	Pro-270, Arg-450, Phe-274
Kaempferol		−18.95	Cys-177, Cys-186, Tyr-486
Moracin A		−19.41	Asp-36, Pro-34, Tyr-79
Astragaloside IV		−15.27	Asp-259
Astragaloside III		−15.06	Ile-268, His-487, Tyr-486
Puerarin	UGT1A9	−15.48	Phe-271
Scopoletin		−20.42	Leu-173
Tiliroside		−10.04	Gly-273, Phe-257
Formononetin		−27.99	Gly-392, Pro-149, Tyr-245
Kaempferol		−23.39	Gly-171, Phe-150, Leu-173
Moracin A		−20.75	Ser-309, Glu-310
Astragaloside IV		−13.01	His-213, Ala-185, Gln-182, Ala-475,Leu-478
Astragaloside III		−15.10	Arg-439, Arg-447, His-444, Phe-271

## Data Availability

The study data are contained within the article.

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
