# Peer review of "Active Constituent of HQS in T2DM Intervention: Efficacy and Mechanistic Insights"

_ijms, 2025, doi:10.3390/ijms26104578_

Round 1
Reviewer 1 Report
Comments and Suggestions for Authors
This study evaluates the therapeutic efficacy and mechanisms of 12 active constituents (AC) from Huang-Qi San (HQS) in treating type 2 diabetes mellitus (T2DM). These constituents include puerarin, formononetin, and astragaloside, among others. They were selected using metabolomics, bioinformatics, molecular docking, and other in silico methodologies. Although it is challenging to analyze and comment on these methods comprehensively, we have to assume that the chosen methods were accurate. The selection of the most active and safest constituents, and the better dosages, based on their effects on T2DM targets, can be considered an important strategy. Using a combination of pure compounds has as advantages, the standardization of the phytomedicine, avoiding problems of sazonality, soils, and environmental effects on the quality of the phytotherapy.
This study evaluates the therapeutic efficacy and mechanisms of twelve active constituents (AC) from Huang-Qi San (HQS) in treating type 2 diabetes mellitus (T2DM). The constituents, which include puerarin, formononetin, and astragaloside, were selected using metabolomics, bioinformatics, molecular docking, and other in silico methodologies. Although a comprehensive analysis and commentary on these methods can be challenging, we must assume that the chosen methods are accurate.
Selecting the most active and safest constituents, as well as determining the optimal dosages based on their effects on T2DM targets, can be considered a crucial strategy. Utilizing a combination of pure compounds offers several advantages, including the standardization of the phytomedicine and the mitigation of issues related to seasonality, soil, and environmental variability that can affect the quality of the phytotherapy.
The study is robust, systematically performed, well-organized, and technically sound. It employs appropriate controls and uses metformin as a standard. It leads to improved results. However, several questions must be addressed before a final recommendation can be made.
- 12 components vs. Metformin. Is metformin adequate for the animals' model? Does it reflect also the side effects of its use? Metformin is the subject of controversial efficacy. Is it a better model? Could the authors have used natural products instead of metformin?
- What are the potential interactions between the 12 chemical components, and do they influence each other's effects? Has the bioavailability of the selected constituents been adequately assessed? What collateral effects are associated with Huang-Qi San (HQS), and how do they impact its overall safety and efficacy?
- Do they act in the same organ? Do they reach the targets?
- Are there any collateral effects related to HQS? It would be good to describe them.
- The obtained results justify the use of a more expensive product?
These are general questions that need to be answered or deserve some explanation.
Minor points:
- A flow chart is necessary. It would clarify the enormous number of experiments performed.
- In the experimental part, the extract was obtained at high temperature (around 90 oC, three times. Does the temperature cause any degradation of the compounds¿
- The preparation of the extracts and the pure substances' combination is not detailed. How about solubility¿ Which solvent or buffer was used to prepare the samples?
- Any problem with the gavage administration?
- Is there a best chemical component?
- Describe the limitations and advantages of your mixture.
- Will this combination have economic appeal?
:
Reviewer 2 Report
Comments and Suggestions for Authors
This article discusses how active constituents in Huang-Qi San (HQS) are responsible for its anti-diabetic effects. The research addresses a scientific issue of public interest and meets publication standards overall. However, there are several aspects that require modification.
Firstly, the method of using insulin to stimulate glucose uptake in liver cells, although capable of measuring the cells' response to insulin signaling, should not be termed an insulin resistance model as it does not involve actual insulin resistance. The authors need to revise their terminology accordingly.
Secondly, while the authors emphasize that the efficacy of several active components is superior to that of HQS, they have not conducted statistical tests to support this conclusion, particularly for the Glucose Tolerance Test (GTT) and Insulin Tolerance Test (ITT). Additionally, it is unclear whether the dosage of AC_H is higher or lower than the dosage of active components in HQS. If the dosage of AC_H is significantly higher, it is not surprising that its effects are better.
Moreover, the results of the study do not appear to address the paper's objectives. There is no data supporting the synergistic effects among different active components in HQS, nor is there evidence that the treatment effects of single active substances are weaker than those of the combined active substances. These limitations need to be reflected in the text.
Reviewer 3 Report
Comments and Suggestions for Authors
This study encompasses a mechanistic study of the active components of Huang Qi San, considering their synergistic properties using in silico, in vitro, and in vivo models, as a potential therapeutic agent for type 2 diabetes. The study is very comprehensive; however, there are some methodological considerations that should be taken into account:
- The animal model used claims to resemble type 2 diabetes; however, how is it possible for severe hyperglycemia of more than 20 mmol/L to coexist with hyperinsulinemia? With such high glucose levels, beta cells would no longer be able to compensate for the existing insulin resistance, resulting in low insulin levels.
- The caloric and nutritional content of the diets is not specified.
- Cervical dislocation is no longer considered a bioethical method of euthanasia.
- Why was insulin used to generate "insulin resistance" in the in vitro model? Wasn't an alternative, such as a type of fat, considered? Why wasn't the metformin group treated with insulin like the experimental treatments?
- Figure 6: It is unclear whether food and water consumption is daily or per rat. Was caloric intake or feed efficiency ratio calculated?
- Glycated hemoglobin should be expressed as a percentage. Hyperinsulinemia cannot occur at levels above 7%.
- To properly analyze insulin tolerance tests, the inverse areas under the curve or glucose disposal constants must be calculated. Glucose disposal rate must be considered. It would be also important to calculate the change from baseline.
Reviewer 4 Report
Comments and Suggestions for Authors
1. The novelty of the research— Comparison of HQS active constituents, the extract, and metformin—is not explicitly declared. The manuscript would be improved by a clear declaration of novelty and specific hypothesis in the abstract and introduction.
2. Despite the fact that metabolomics and molecular docking are performed, mechanistic interpretation is shallow. The manuscript should better integrate identified pathways and binding affinities with in vivo and in vitro pharmacodynamics observed.
3. Synergistic interactions among active components are stated, but there is no statistical synergy analysis (e.g., combination index, Bliss independence). This takes away from a key paper claim.
4. Using insulin-resistant HepG2 cells only decreases generalizability. Either add a second cell line (e.g., 3T3-L1 or L6 myotubes) or explicitly mention this limitation in the discussion.
5. Molecular docking provides theoretical binding scores, but no validation experiments (e.g., enzymatic activity assays, Western blot) are presented. This lowers confidence in binding conclusions.
6. Many sentences, especially in the discussion, summarize previously reported results. Streamline writing by reducing redundancy and strengthening paragraph transitions.
7. Though metabolites and pathways are given, the biological relevance of changes (i.e., how cucurbitacin and palmitic acid are linked to T2DM development) is not discussed in depth. This limits reader insight.
8. The results often use "significantly increased/decreased" language but lack exact values, fold changes, or effect sizes needed for Q1-level impact.
9.Figures (particularly 10–12) need to be more clearly labeled, with larger fonts, consistent coloring, and improved legends. Include schematic models that summarize mechanisms where possible.
10. Some of the citations are old. For a Q1 journal publication, the expectation is that at least 50% of references are within the last 3 years, particularly for TCM pharmacology and diabetes mechanisms.
11. Translational value: the paper does not contain any pharmacokinetics (e.g., bioavailability, plasma levels) of the active components.
12. Docking is studied on UGT enzymes only. Metabolomics data demonstrate involvement of pathways including fatty acid metabolism. Additional docking with relevant targets, e.g. ACC, FASN, should be performed.
13. More details on statistical methods used, for example in multivariate metabolomics or grey correlation analysis, would be necessary, including specific packages used, thresholds applied, and multiple comparison correction.
14. The paper combines the eight actives in equal parts, yet there is neither explanation nor experimental rationale for doing so. Would another proportion be more effective?
15. The abstract has no numbers and uses vague terms like "significant" and "outperforming." A more formal format (Background – Methods – Results – Conclusion) will make it clearer and more effective.
English laguage is ok, but some proffiecient reading would improve quality.
Reviewer 5 Report
Comments and Suggestions for Authors
This is a good basic science study and provides some insight into potential Chinese remedy with regards to diabetics. My main suggestion would be to consider having materials and methods placed before the discussion portion. Aside from that including a conclusion section at the end of the manuscript in addition to the one included in the abstract would be advised. Limitations and strengths of the study can be mentioned as well for the readers.
Author Response
Comment: This is a good basic science study and provides some insight into potential Chinese remedy with regards to diabetics. My main suggestion would be to consider having materials and methods placed before the discussion portion. Aside from that including a conclusion section at the end of the manuscript in addition to the one included in the abstract would be advised. Limitations and strengths of the study can be mentioned as well for the readers.
Response: Dear Reviewer, we sincerely appreciate your thorough review and constructive suggestions regarding our manuscript. The structural sequence of the main text you proposed has been composed in accordance with the journal's template and cannot be altered. Additionally, in response to your suggestion, we have included a conclusion section at the end of the manuscript to more effectively summarize the strengths and limitations of the manuscript. Once again, thank you for your recognition and invaluable suggestions. We look forward to your further feedback.
Round 2
Reviewer 4 Report
Comments and Suggestions for Authors
I don't have any further comments. The paper can be considered for publication.